# Hydrological regime shifts in Sahelian watersheds: an investigation with a simple dynamical model driven by annual precipitation

Erwan Le Roux[1], Valentin Wendling[2], Gérémy Panthou[3], Océane Dubas[3], Jean-Pierre Vandervaere[3], Basile Hector[3], Guillaume Favreau[3], Jean-Martial Cohard[3], Caroline Pierre[4], Luc Descroix[5], Eric Mougin[6], Manuela Grippa[6], Laurent Kergoat[6], Jérôme Demarty[7], Nathalie Rouche[7], Jordi Etchanchu[7], and Christophe Peugeot[7]

[1]IMT Atlantique, Lab-STICC, UMR CNRS 6285, 29238, Brest, France
[2]HydroSciences Montpellier (Univ. Montpellier, IMT Mines Ales, CNRS, IRD) Ales, France
[3]Institut des Géosciences de l'Environnement (Univ. Grenoble Alpes, INRAE, CNRS, IRD, Grenoble INP), Grenoble, France
[4]Institut d'Ecologie et des Sciences de l'Environnement de Paris (CNRS, Sorbonne Univ., Univ Paris Est Creteil, IRD, INRAE, Univ. de Paris) Paris, France
[5]Patrimoines locaux, Environnement et Globalisation (MNHN, IRD, CNRS), Paris, France
[6]Géosciences Environnement Toulouse (CNRS, IRD, UPS, CNES) Toulouse, France
[7]HydroSciences Montpellier (IRD, Univ. Montpellier, CNRS) Montpellier, France

**Correspondence:** Christophe Peugeot (christophe.peugeot@ird.fr)

**Abstract.** The Sahel, the semi-arid fringe south of the Sahara, experienced severe meteorological droughts in the '70s-'80s. During and after these droughts, watersheds in the Central Sahel have experienced an increase in the annual runoff coefficient (annual runoff normalized by annual precipitation). We hypothesize that these increases correspond to regime shifts. To investigate the timing of these regime shifts, we introduce a lumped model that represents feedbacks between soil, water and vegetation at the watershed scale and the annual time step. This model relies on runoff coefficient as a constraint for the state variable and precipitation as unique external forcing. Four watersheds (Gorouol, Dargol, Nakanbé and Sirba), with pluri-decennial observations ('50s-2010s), are modeled. For each watershed, one million parameterizations of this model are sampled and run, and an ensemble of one thousand best parameterizations is selected based on observed runoff coefficients. Our results show that this model can reproduce the trend of runoff coefficients. For all watersheds, almost all selected parameterizations from the ensemble are bistable. We define two alternative runoff coefficient regimes (a low and a high regime) by splitting with a threshold the bifurcation diagram of bistable parameterizations. Most selected parameterizations undergo regime shifts: simulated runoff coefficients belong to the low regime in 1965 and to the high regime in 2014. Finally, we find that the year of the regime shift, defined as the year when the number of regime shifts is maximized, was 1971, 1972, 1973, 1983 for the Gorouol, Nakanbé, Dargol and Sirba watershed, respectively. These results were obtained with a parsimonious model which deliberately neglects fine-scale processes of Sahelian hydrology. It would therefore be wise to supplement this analysis with other models — with varying levels of complexity — that also allow regime shifting. Overall, this article proposes simple ideas toward improving the modelling and characterization of hydrological regime shifts.

# 1 Introduction

Complex dynamical systems (ecosystems, climate system) can have, for certain external conditions, several attractors (stable states) towards which the system state converges depending on its initial value (Scheffer et al., 2001). The set of initial values converging to the same attractor defines an attraction basin. Classically, one regime is associated with each attraction basin (Mathias et al., 2024). A regime shift, i.e. the transition from one regime to an alternative regime, corresponds to the crossing of a tipping point: a critical value beyond which a system switches to the alternative regime, often abruptly and/or irreversibly (IPCC, 2023). Tipping points are often associated with critical thresholds in the external condition of the dynamical system (Armstrong McKay et al., 2022). However, tipping points can also refer to a critical level of noise in the state space or a critical rate of change in the external condition (Lenton, 2011; Ashwin et al., 2012).

In hydrological science, better accounting for regime shifts remains a challenge (Blöschl et al., 2019; van Hateren et al., 2023) as commonly used hydrological models poorly account for such potential abrupt and/or irreversible changes (Saft et al., 2016; Fowler et al., 2016, 2020, 2022a). Some approaches use statistical methods to analyze regime shifts (Zipper et al., 2022; Fowler et al., 2022b; Rahimi et al., 2023; Goswami et al., 2024). In particular, several studies have highlighted a shift of watersheds toward lower streamflow after a drought due to shifts in the relationships between surface water and groundwater (Saft et al., 2015; Peterson et al., 2021; Fowler et al., 2022a; Liu et al., 2023). Other approaches rely on dynamical models to analyze potential regime shifts. In Australia, Anderies (2005) and Anderies et al. (2006) extract stable states from a dynamical model applied to a catchment, Peterson et al. (2012) extract attractors, repulsors and attraction basins for a simple lumped groundwater model, while Peterson and Western (2014) investigate multiple hydrological attractors with a semi-distributed hillslope eco-hydrological model. In Northern Mali, Wendling et al. (2019) model a shift from a high-vegetation/low-runoff regime to a low-vegetation/high-runoff regime. In Germany, Dijkstra et al. (2019) rely on an idealized model to show that a shift from a low to a high sediment concentration regime may occur in the Ems River assuming low river discharge and increasing channel depth. Eco-hydrological regime shifts have also been studied with spatial models of dryland vegetation (Van Nes and Scheffer, 2005; Mayor et al., 2019; Kéfi et al., 2024).

This study focuses on the Sahelian hydrological paradox (Mahe et al., 2005; Descroix et al., 2009). In several watersheds in the Central Sahel (Mali, Burkina Faso, Niger), the annual runoff coefficients (annual runoff normalized by annual precipitation) increased during the major meteorological droughts in the '70s-'80s and surprisingly kept increasing after them. This paradox is attributed to the hydrological effect of land clearing (Leblanc et al., 2008; Favreau et al., 2009) which favors soil crusting and thus Hortonian surface runoff (Horton, 1933), the dominant runoff generation process in the region (Casenave and Valentin, 1992). In the northernmost areas where rainfed agriculture is not possible, the paradox is attributed to drought-induced vegetation dieback, which results in similar effects on soil properties (Hiernaux et al., 2009; Gardelle et al., 2010; Gal et al., 2017).

Multiple lines of evidence suggest that this Sahelian hydrological paradox corresponds to a hydrological regime shift. These evidences include a cessation of the disturbance, i.e. the drought that ended in the mid- 1990s (Panthou et al., 2018), a shift in annual runoff coefficient, and an evidence of non-recovery (Descroix et al., 2009, 2018).

Here, we focus on a complementary line of evidence based on a dynamical model, where the occurrence of a regime shift is a starting assumption. Our objective is to better characterize these regime shifts by analyzing their timing. Other works have previously identified the timing of shifts, where regimes are identified statistically (Fig. S19 and S20 of Peterson et al. 2021). By contrast, in this study, regimes are defined using attractors from the dynamical model.

Our scientific approach is structured in three phases. First, for four Sahelian watershed, where major hydrological changes have been documented between the 1950s to mid-2010s (Gal et al., 2017; Descroix et al., 2009, 2018; Sauzedde et al., 2025), we assume that the observed hydrological changes (Sahelian paradox) correspond to a regime shift from a low to a high runoff coefficient regime. Then, we propose a dynamical model that can account for a regime shift, and evaluate if it can reproduce the observed trend in runoff coefficients. Finally, based on this model, we ask: when did these shifts occur ?

In our methodology, we propose several novel contributions for the second and third phases of this approach. The second phase requires a dynamical model that can account for a hydrological regime shift of Sahelian watersheds. However, to our knowledge, such a model does not yet exist. To fill this gap, we build on previous work in ecological modelling with multiple attractors (Holling, 1973; May, 1977; van Nes et al., 2014; Wendling et al., 2019) to develop a lumped dynamical model that can simulate these regime shifts. Following the parsimony principle, we deliberately chose to develop a minimal model capable of reproducing the first-order processes of such complex eco-hydrological systems (Scheffer et al., 2001; Sivapalan, 2018) because comprehensive datasets, required to inform more complex models, are not available in this region for the investigated period. Specifically, this model simulates annual runoff coefficient at the watershed scale and the annual time step, using annual precipitation as unique external forcing. This model incorporates a new formalism capable of representing feedbacks between vegetation dynamics and runoff generation (soil crusting, erosion processes and growth and death of vegetation patches). The model is constrained by the observed runoff coefficients in order to ensure realistic simulations. The third phase of our approach, that consists in assessing the timing of regime shift, requires to quantitatively identify a regime shift. For this purpose, we propose a definition of regimes that splits the state space into two sub-spaces (a low and a high regime) and develop a method to identify when a transition between these two regimes occurs.

This paper is organized as follows. Section 2 presents our data. Section 3 develops our methodology. Results, discussions and conclusions are introduced in Sects. 4, 5 and 6, respectively.

## 2  Data

### 2.1  Sahelian watersheds

The four considered watersheds (Gorouol, 23200 km$^2$; Dargol, 7350 km$^2$; Sirba, 40300 km$^2$; Nakanbé, 21800 km$^2$), located in
central Sahel (Fig. 1), are selected on the basis of the availability of long-term observations from the 1950s to the mid-2010s.
The first three are tributaries of the Niger River, and the last one feeds the Volta River. The climate of this semi-arid region
is governed by the West African monsoon (Redelsperger et al., 2006; Lafore et al., 2011). Seasonal precipitation is provided
by mesoscale convective systems between June and September. Intermittent rivers are supplied by surface runoff generated
on hillslope during rainstorms. The study area is essentially rural: the north is dominated by natural pastures grazed by cattle,
while the south contains a mosaic of savannahs, rainfed agricultural plots and fallows, where scattered shrub and tree species
persist (Tucker et al., 2023).

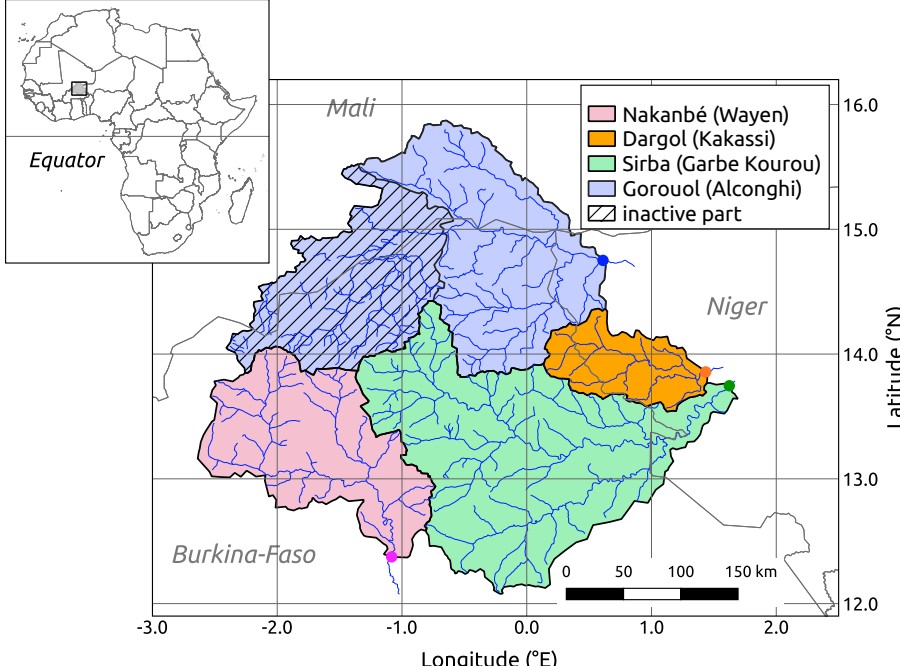

**Figure 1.** Location of the study area in Sub-Saharan Africa, and map of the four watersheds. The grey lines are the country borders. The
large dots show the watershed outlet. The names of corresponding streamflow stations are in brackets in the legend. The upper Gorouol basin
was inactive between the 1950s and the 2010s and does not contribute to the streamflow. Only the active sub-basin is considered.

## 2.2 Annual precipitation

Annual precipitation over each watershed are computed based on Panthou et al. (2014, 2018) First, for each gauge station of the watershed, annual precipitation series are obtained by summing daily precipitation for each year. Then, annual precipitation means and anomalies are both interpolated using two kriging methods. Finally, annual precipitation of the watershed is computed as the sum of these two interpolated fields (means and anomalies) averaged spatially over the watershed.

Figure 2a presents annual precipitation over four watersheds for the period 1956-2014. Past annual precipitation can be grouped in three main periods: wet during the '50s-'60s, followed by a prolonged period with some severe droughts which ended up in the mid-'90s, and since then a period with a strong inter-annual variability where annual precipitation roughly equals the mean of the whole period (Le Barbé et al., 2002).

## 2.3 Annual runoff coefficient

The annual runoff coefficient $K$ (-), ranging between 0 and 1, is equal to the annual watershed outflow $V$ (m$^3$) per unit of watershed area $A$ (m$^2$), normalized by the annual precipitation $P$ (mm): $K = \frac{V}{A \times P}$. Annual watershed outflow $V$ (m$^3$) is calculated from the mean daily discharge $Q$ (m$^3$ s$^{-1}$), over the hydrological year, starting on the first of March. For the Nakanbé watershed, we use discharges corrected for the effect of dams built in recent decades (Gbohoui et al., 2021). For the three other watersheds, discharges are merged from two databases: SIEREM (Boyer et al., 2006) and ADHI (Tramblay et al., 2021). We exclude years with more than 13 days of missing consecutive daily discharge values during the wet season (June-September). Otherwise, missing values are set to zero in the dry season, and linearly interpolated in the wet season. A sensitivity study (not shown) confirms that this interpolation has a limited effect on the mean annual discharge (bias < 10%).

Figure 2b presents the annual runoff coefficient series for each watershed. All watersheds have experienced a similar evolution toward higher annual runoff coefficients although the Northern watersheds display higher coefficients.

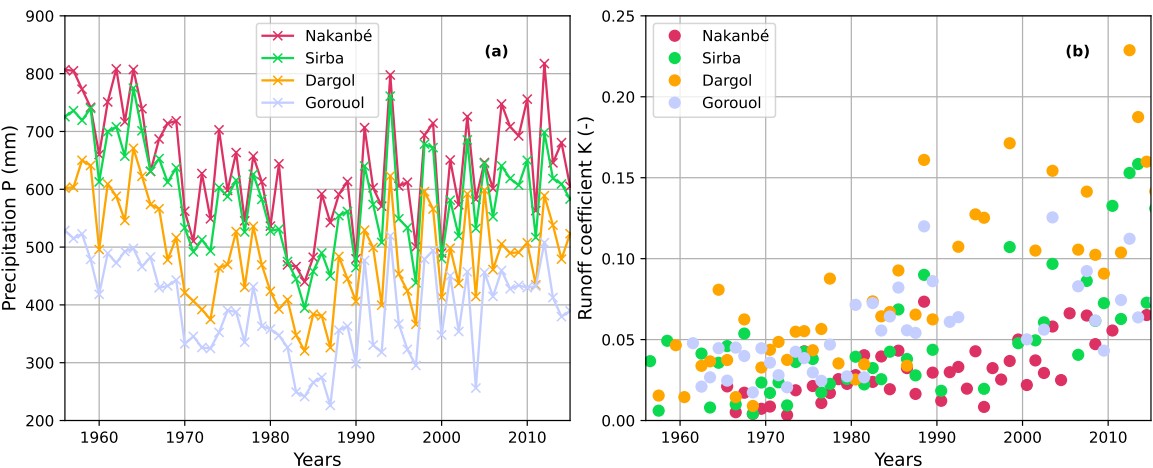

**Figure 2.** Observations of (left) annual precipitation and (right) annual runoff coefficient for the four watersheds between 1956 and 2014.

## 3 Methodology

### 3.1 Dynamical model

In this article, variables are denoted with uppercase letters, whereas parameters are written with lowercase letters. Furthermore, in this study, most Sahelian watersheds have sub-domains that do not contribute to runoff at the outlet due to their endorheic nature. In order to account for this partial contribution, our model only focuses on the contributing areas. Therefore, all physical variables are defined at annual time step and at the scale of the watershed areas contributing to runoff. For instance, the runoff coefficient $K$ refers to the annual runoff coefficient for the areas of the watershed contributing to runoff.

The proposed dynamical model is an extension of the hillslope-scale model proposed by Wendling et al. (2019), which reproduced a hydrological shift. Our lumped dynamical model simulates the runoff coefficient $K$. This parsimonious model has a single external forcing, the precipitation $P$ (mm), and a single state variable, the water holding strength $S$ (-). Intuitively, $S$ represents all physical mechanisms that drive water retention within the watershed including soil infiltration properties associated with vegetation dynamics, seepage losses or conversely runoff connectivity. $S$ ranges from 0 to 1. The higher is $S$, the lower is $K$. In the model, the variations in $S$ are assumed to be similar to those of a vegetation cover: $S$ increases and decreases as a function of $I$ (mm), an indicator of wetness representing the potential water for the vegetation, i.e. precipitation minus runoff.

Changes in the water holding strength $S$ are modulated by a feedback loop (Fig. 3). The value of $S$ impacts directly the proportion of outflow water $K$, that indirectly affects the indicator of wetness $I$, which finally drives the growth and decays of $S$. These assumptions are based on feedback mechanisms frequently observed in drylands worldwide (Turnbull et al., 2012). Wet conditions favor vegetation development, which in turn favors infiltration and thus vegetation growth (i.e. high $S$ values). Conversely, dry conditions are detrimental to vegetation, favoring bare soil and consequently low infiltration (i.e. low $S$ values).

Mathematically, changes in $S$ are prescribed using a differential equation, where the time derivative $\frac{dS}{dt}$ is denoted as $\dot{S}$. The computation of $\dot{S}$ can be decomposed into three steps (Eq. 1, Eq. 2, Eq. 3), illustrated in Figure 3, and detailed below.

First, we define the runoff coefficient $K$ as a function of the water holding strength $S$ and the precipitation $P$:

$$K = k_{max} \cdot \left( \frac{P^a}{P^a + C^a} \right)^b \text{ with } C = c_{min} + S \times (c_{max} - c_{min}) \tag{1}$$

where $C$ (mm) is the water holding capacity; $a$ (-), $b$ (-), $k_{max}$ (-), $c_{min}$ (mm), $c_{max}$ (mm) are positive parameters (Tab. 1). Equation 1 is a fully derivable variant (well adapted to ODE solving) of the popular SCS model (Mockus, 1972), and of the equation used by (Massuel et al., 2011) to simulate runoff in the Sahel. It is also very similar to the formalism used by (Anderies 2005, Eq. 20) to represent the dependence of runoff to precipitation and to the watershed-scale water retention capacity, which is analogous to our variable $S$. Equation 1 is an S-shape function that controls the precipitation-runoff relationship: no runoff is produced below a certain precipitation amount, the runoff ratio varies only little for heavy precipitation, and it increases roughly linearly for intermediate precipitation. The runoff coefficient $K$ is maximized when precipitation is substantially higher than the water holding capacity ($P \gg C$), i.e. when $S$ is close to 0. Inversely, the runoff coefficient $K$ is close to zero if $C \gg P$, i.e.

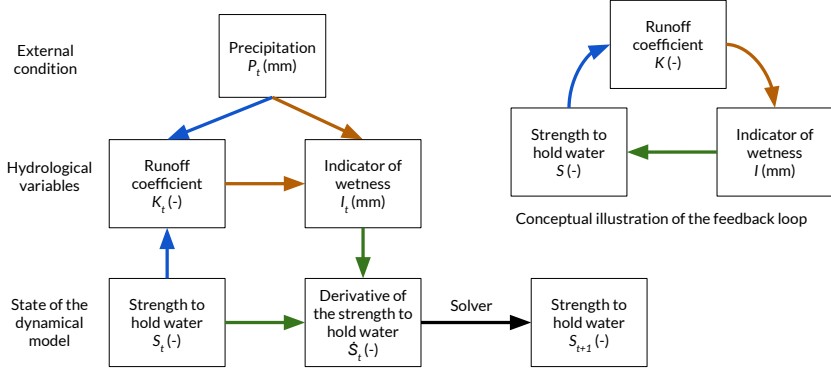

**Figure 3.** At every time $t$, the derivative of the water holding strength $\dot{S}_t$ is computed in three steps: i) the runoff coefficient $K_t$ is derived from the water holding strength $S_t$ and the precipitation $P_t$ (Eq. 1), ii) the indicator of wetness $I_t$ is determined by the runoff coefficient $K_t$ and the precipitation $P_t$ (Eq. 2), iii) the derivative $\dot{S}_t$ is calculated with the water holding strength $S_t$ and the indicator of wetness $I_t$ (Eq. 3).

when $S$ is close to 1. The parameters $a$ and $b$ are shape factors allowing the model to adapt to a wide range of watersheds. This

equation, although not properly physically-based, proposes a representation consistent with known hydrological processes.

In the second step, the indicator of wetness $I$ (mm) is derived from the precipitation $P$ and the runoff coefficient $K$:

$$I = P \cdot (1 - K) \tag{2}$$

Finally, we further assume that the derivative of the water holding strength $\dot{S}$ can be calculated from $S$ and $I$:

$$\dot{S} = \overbrace{r_g \cdot \frac{I}{I + i_g} \cdot S \cdot (1 - S)}^{(1)} - \overbrace{r_d \cdot \frac{i_d}{I + i_d} \cdot S}^{(2)} + \overbrace{\mu \cdot (1 - S)}^{(3)} \tag{3}$$

where $r_g$ (year$^{-1}$), $i_g$ (mm), $r_d$ (year$^{-1}$), $i_d$ (mm) and $\mu$ (year$^{-1}$) are positive parameters. Equation 3 contains a growth term (1) and a decay term (2), whose rates ($r_g$ and $r_d$) are modulated by $I$. The balance between these terms depends on $S$ and $I$, and determines the sign of the derivative of $S$, hence its variation. The third term of Eq. 3 is small and only prevents the unrealistic convergence to $S = 0$. It can be interpreted as the representation of all the physical processes not included in the model which increase the water holding strength such as sand deposits by wind and vegetation growth due to seeds imported.

In general, this third term $\mu \cdot (1 - S)$ remains negligible as compared to the other terms of Eq. 3, because the range of values of $\mu$ (Tab. 1) are 2 to 3 orders of magnitude lower that the growth and decay rates $r_g$ and $r_d$. The only exception is when $S$ becomes close to 0: in this case terms 1 and 2 in Eq. 3 are also close to 0, and $\dot{S}$ is close to $\mu$ (always positive), avoiding the trajectory of $S$ to remain stuck in 0.

## 3.2 Selection of best parameterizations for the model

For each watershed, the selection of best parameterizations rely on a comparison between observed runoff coefficients $K_{observed}$ with simulated runoff coefficients $K_{simulated}$ at the watershed scale. The watershed-scale simulated runoff coefficient $K_{simulated}$ can be computed from $K$, the runoff coefficient of the areas contributing to runoff (Eq. 1), as follows:

$$K_{simulated} = f \times K \tag{4}$$

where $f$ (-) is the fraction of the watershed (between 0 and 1) that actually contributes to the watershed outlet.

Let $\boldsymbol{\theta} = \{k_{max}, a, b, c_{min}, c_{max}, r_g, r_d, i_g, i_d, \mu, f\}$ denote a parameterization of the dynamical model, i.e. a set of parameters for the equations (Eq. 1, Eq. 2, Eq. 3, Eq. 4). For each parameter, values (or ranges of values) are shown in Table 1.

| Parameter | Description | Unit | Value/Range |
|-----------|-------------|------|-------------|
| $r_g$ | Growth rate | $y^{-1}$ | [0.2, 2.0] |
| $r_d$ | Decay rate | $y^{-1}$ | [0.5, 5] |
| $i_g$ | Half-saturation constant for growth | mm | [150, 700] |
| $i_d$ | Half-decay constant for decay | mm | [10, 200] |
| $\mu$ | Minimum growth rate | $y^{-1}$ | [2e-3, 5e-3] |
| $c_{min}$ | Minimum value of water holding capacity | mm | [0, 140] |
| $c_{max}$ | Maximum value of water holding capacity | mm | [600, 800] |
| $k_{max}$ | Maximum runoff coefficient | - | 0.9 |
| $a$ | Shape parameter (steepness) | - | 1.5 |
| $b$ | Scale parameter (inflection point) | - | 8.0 |
| $f$ | Land fraction contributing to the discharge outlet | - | [0.1, 1] |

Table 1. Parameters of the dynamical model: name, description, unit and value or range of values (adjusted with preliminary tests).

Selecting a single best parameterization $\boldsymbol{\theta}^*$ can be prone to equifinality, i.e. different parameterizations lead to similar results (Beven, 2006). To circumvent this issue, we adopt a three-step approach to select an ensemble of relevant parameterizations:

1. $10^6$ parameterizations are sampled from a Latin hypercube sampling, using a uniform distribution over a range of pa-
rameter values (Table 1). These ranges were adjusted through preliminary tests. Note that $a$, $b$ and $k_{max}$ are fixed.

2. For each parameterization, using observed precipitations as a forcing, the trajectory of water holding strength $S$ and runoff coefficient $K$ are solved numerically by coupling 1, 2 and 3 with a wrapper of the Fortran solver from ODEPACK (Hindmarsh, 1983). The trajectory of simulated runoff coefficient $K_{simulated}$ is then inferred from $K$ using Eq. 4.

3. Following (Wendling et al., 2019), the best 1000 parameterizations are selected based on the lowest values of the root
mean squared error (a classical objective function) between $K_{observed}$ and $K_{simulated}$. This ensemble of the 1000 best parameterizations is denoted as $\boldsymbol{\Theta}^* = \{\boldsymbol{\theta}^{(1)}, ..., \boldsymbol{\theta}^{(1000)}\}$.

### 3.3 Monostable and bistable parameterizations

For each parameterization $\theta$ of the ensemble $\Theta^*$, we compute its bifurcation diagram (Fig. 4). This diagram is a graph that associates each forcing value (precipitation $P$) with its fixed points (attractors, repeller), i.e. values $S$ such that $\dot{S} = 0$.

Here fixed points are calculated using a continuation method that finds the isoline defined by $\dot{S} = 0$, starting from an initial solution. In practice, we rely on the "pycont-lint" Python package, which is dedicated to numerical bifurcation analysis. We begin by setting the state value close to $0$ ($S = 0.01$) for a very low precipitation ($P = 0.1$), and stop the continuation method when the isoline exceeds 4000 mm of precipitation. The precipitation range explored (until 4000 mm) is on purpose larger than the actual precipitation range (Sect. 2.2) in order to assess the sensitivity of the continuation method.

We define that a parameterization $\theta$ is monostable if it has a single fixed point (an attractor) for every precipitation value between 1 and 4000 mm, and bistable if it has several fixed points for at least one precipitation value. In this case, the continuation algorithm provides both the stable (attractor) and unstable (repeller) fixed points. Figure 4 illustrates the bifurcation diagram for a monostable parametrization (Fig. 4 **a**), and a bistable parametrization (Fig. 4 **b**). In the bistable case, for any precipitation in the range 700 to 1500 mm, three $S$ values (two attractors, one repeller) can be reached asymptotically depending on the initial state value. We denote as lower and upper branch the lines formed by the lower and upper attractors, respectively.

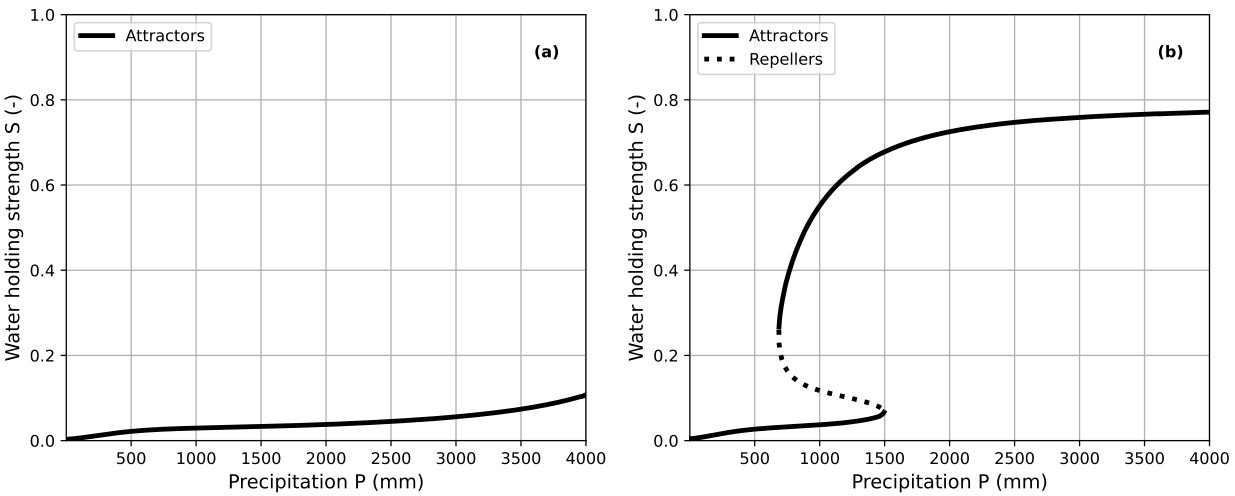

**Figure 4.** Bifurcation diagram of a **(a)** monostable **(b)** bistable parameterization $\theta$ for precipitation $P$ between 1 mm and 4000 mm.

### 3.4 Regime definition

A bifurcation diagram is a useful tool to define regimes. For monostable parameterization (Fig. 4 **a**), there is a unique regime, and changes in the state variable $S$ or in the runoff coefficient $K$ adapt to changes in precipitation in a reversible way. For bistable parameterizations, regimes and regime shifts require more subtle definitions, that we describe below.

Regimes and regime shift, i.e. the transition from one regime to the alternative regime, are often conceptualized with state values that remain equal to attractors (Scheffer et al., 2001). However, many systems (including ours) have transient dynamics, where state values remain far from attractors (Hastings et al., 2018). Here we present two definitions of regimes suited for such a transient case. Both definitions split the bifurcation diagram into two regions: the "Low runoff coefficient regime" and the "High runoff coefficient regime", that correspond to a high water holding strength and low water holding strength, respectively.

Classically, a regime is associated with an attraction basin, i.e. the set of initial values converging to the same branch of attractors (Fig. 5 **a**). One drawback of this classical definition is that a state $S$ cannot be directly classified into one regime, because the attraction basin also depends on the precipitation $P$. In other words, the same state $S$ can switch regimes following changes in precipitation. For instance, $S = 0.2$ corresponds to the "Low runoff coefficient regime" for $P = 750$ mm, and to the "High runoff coefficient regime" for $P = 600$ mm.

We introduce an alternative definition of regimes based on a norm (Mathias et al., 2024). Visually, regimes are separated by a threshold (Fig. 5 **b**). This definition is more practical as any state value $S$ can be directly classified into the low or high regime by comparing it with the threshold. Specifically, $S$ is in the "High runoff coefficient regime" if $S < \diamondsuit_S = \frac{\blacktriangledown_S + \blacktriangle_S}{2}$ while it is in the "Low runoff coefficient regime" if $S \geq \diamondsuit_S = \frac{\blacktriangledown_S + \blacktriangle_S}{2}$. For a given bistable parameterization $\boldsymbol{\theta}$, $\blacktriangle_S$ and $\blacktriangledown_S$ are the water holding strength for the highest attractor of the lower branch and for the lowest attractor of the upper branch, respectively (Fig.

5 **b**). The associated precipitation values are denoted as $\blacktriangle_P$ and $\blacktriangledown_P$.

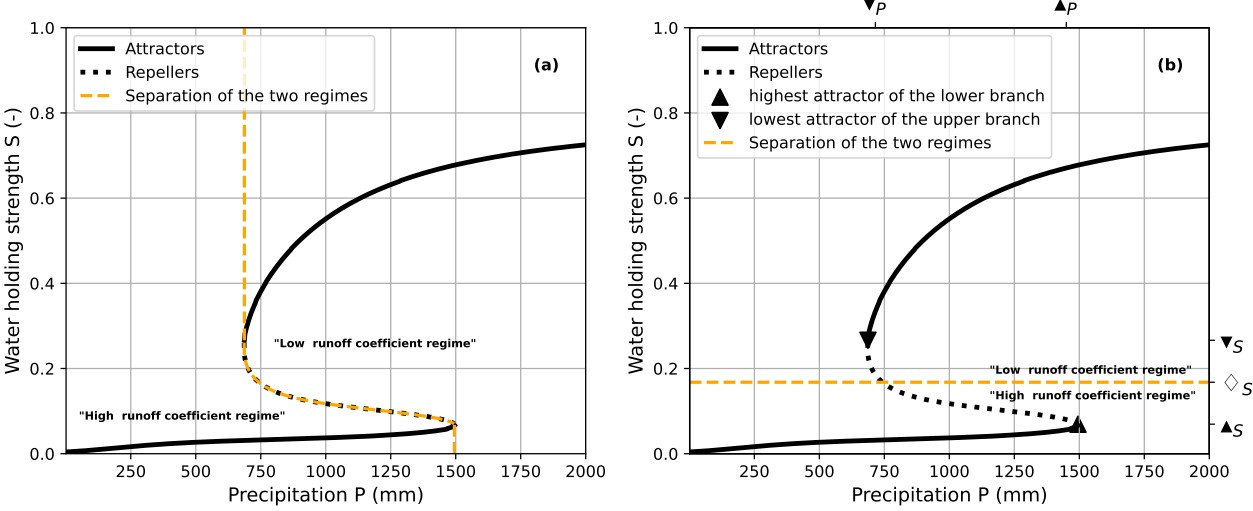

**Figure 5.** Bifurcation diagram of a bistable parameterization $\boldsymbol{\theta}$ for precipitation $P$ between 1 mm and 2000 mm (for readability). We illustrate (a) regimes based on attraction basin (b) regimes based on a threshold $\diamondsuit_S$, where $\diamondsuit_S$ equals the mean of $\blacktriangle_S$ and $\blacktriangledown_S$.

# 4 Results

## 4.1 Ensemble simulations

For each watershed, we study the ensemble $\Theta^*$ made of the best 1000 parameterizations (Sect. 3.2). Figure 6 shows the ensemble of simulated runoff coefficients (Fig. 6 **a,b**) and water holding strength (Fig. 6 **c, d**). The starting year of the trajectories differs between each watershed as the initial state value $S$ is computed with the first observed runoff coefficient and Eq. 1. The Sirba and the Dargol watersheds are initialized in 1956 and 1957, respectively (Fig. 6 **a, c**), whereas the Gorouol and the Nakanbé watersheds are initialized in 1961 and 1965, respectively (Fig. 6 **b, d**).

In Figure 6, we observe that until 2014 the four watersheds have increasing trends of runoff coefficients (and decreasing trends of water holding strength, consistently with Eq. 1). This confirms that this simple dynamical model can reproduce the first-order dynamics (the trend) of the observed runoff coefficient. We note that the ensemble mean water holding strength reaches a plateau around the year 1990 for the Gorouol watershed, and around the year 2000 for the Dargol watershed. A more detailed quantitative assessment of the fit is presented in App. A.

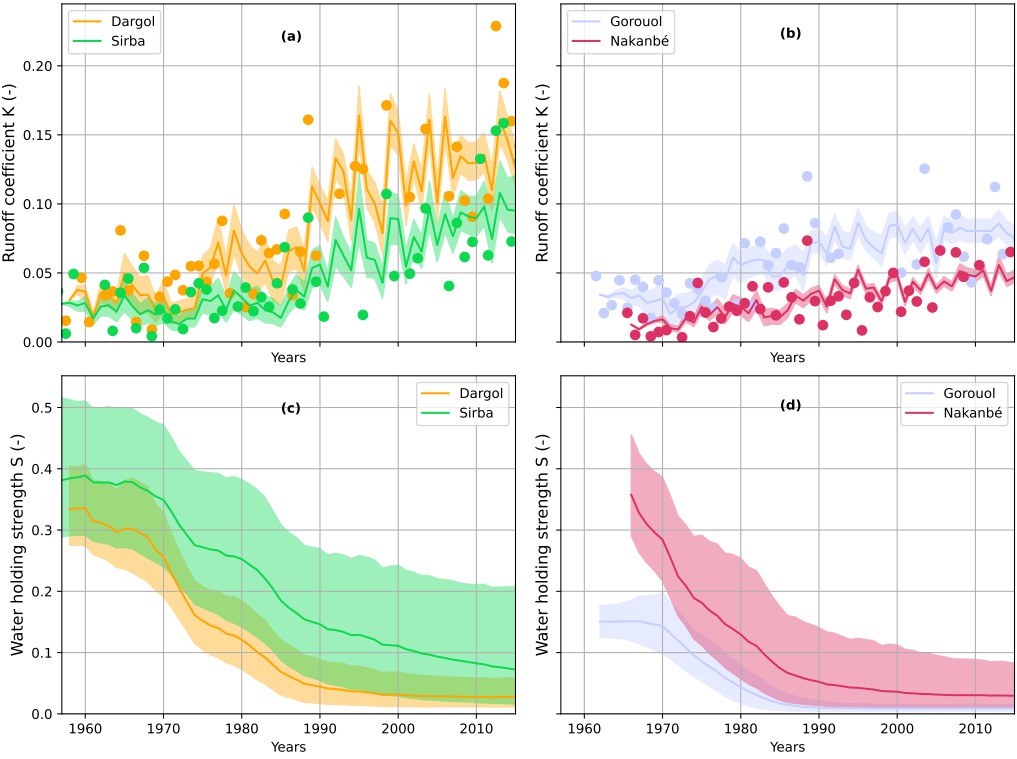

**Figure 6.** Trajectories of the **(a,b)** runoff coefficient and the corresponding **(c,d)** water holding strength for the four watersheds over the period 1956-2014. The colored line emphasizes the ensemble mean, the colored shaded area highlights the range between quantile 5% and quantile 95% of the ensemble. In the Figure **(a,b)** colored dots designate the runoff coefficient observations, as shown in Fig. 2.

### 4.2 Bistable parameterizations

A bifurcation diagram is computed for each parameterization of the ensemble. Bistable parameterizations represent more than
90% of the ensemble for all the watersheds, whereas monostable parameterizations amount to 8.1%, 6.4%, 4.4% and 0% for
the Dargol, Nakanbé, Sirba, and Gorouol watershed, respectively. For these bistable parameterizations, Figure 7 shows the
distribution of $\blacktriangledown = (\blacktriangledown_P, \blacktriangledown_S)$ the lowest attractor of the upper branch and $\blacktriangle = (\blacktriangle_P, \blacktriangle_S)$ the highest attractor of the lower
branch. $\blacktriangledown_P$ and $\blacktriangle_P$ are bifurcation-induced tipping points, i.e. where small changes in annual precipitation can cause a shift to
the alternative regime. In general, we find that the distribution of $\blacktriangledown_P$, a tipping point for the shift to the high runoff coefficient
regime, is less spread than the distribution of $\blacktriangle_P$, a tipping point for the inverse transition. This is because this inverse transition
(the decrease of runoff coefficient) was not observed over the study period, and thus $\blacktriangle_P$ can hardly be constrained. Lastly, we
observe that the drier the watershed (box plots in Figure 7), the more the distribution of $\blacktriangledown_P$ shifts towards lower precipitation
values, and the narrower the range of variation of these values becomes.

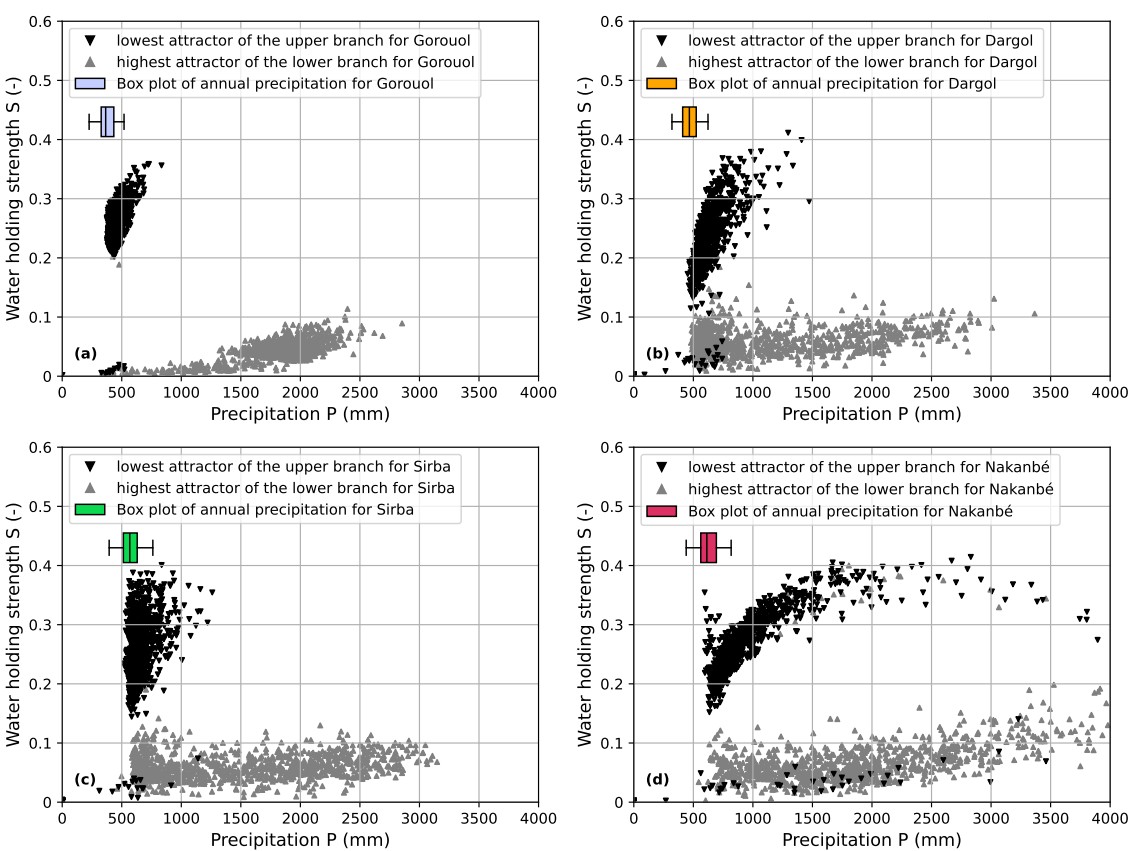

**Figure 7.** Distribution of $\blacktriangledown$ the lowest attractor of the upper branch (in black), and $\blacktriangle$ the highest attractor of the lower branch (in the color
of the watershed) of every bistable parameterization for each watershed. Each box plot (minimum, quantile 0.25, median, quantile 0.75,
maximum) shows the distribution of annual precipitation over the period 1965-2014. For readability, the y-axis was set between 0 and 0.6.

## 4.3 Regime shift

We analyze regime shifts between 1965 and 2014 (common simulation period for all watersheds) by computing, each year, the percentage of selected parameterizations (from the ensemble) that are the "High runoff coefficient regime".

When regimes are defined with attraction basins (Figure 8 **a**), this percentage is non-null in 1965 and increases afterwards for all watersheds, even though the four curves are far from smooth. This is most likely because this definition of regimes is sensitive to the variability in precipitation (Sect. 3.4) which results in a lack of robustness for the identification of regime shift.

When regimes are defined with a threshold (Fig. 8 **b**), all selected parameterizations are in the "Low runoff coefficient regime" in 1965 for the Dargol, Sirba, and Nakanbé watersheds. For the Gorouol watershed, 40% of the selected parameterizations are already in the "High runoff coefficient regime" in 1965. At the end of the simulation period in 2014, more than $80\%$ of selected parameterizations are in a "High runoff coefficient regime" for all watersheds. Thus, most selected parameterizations underwent a regime shift between 1965 and 2014.

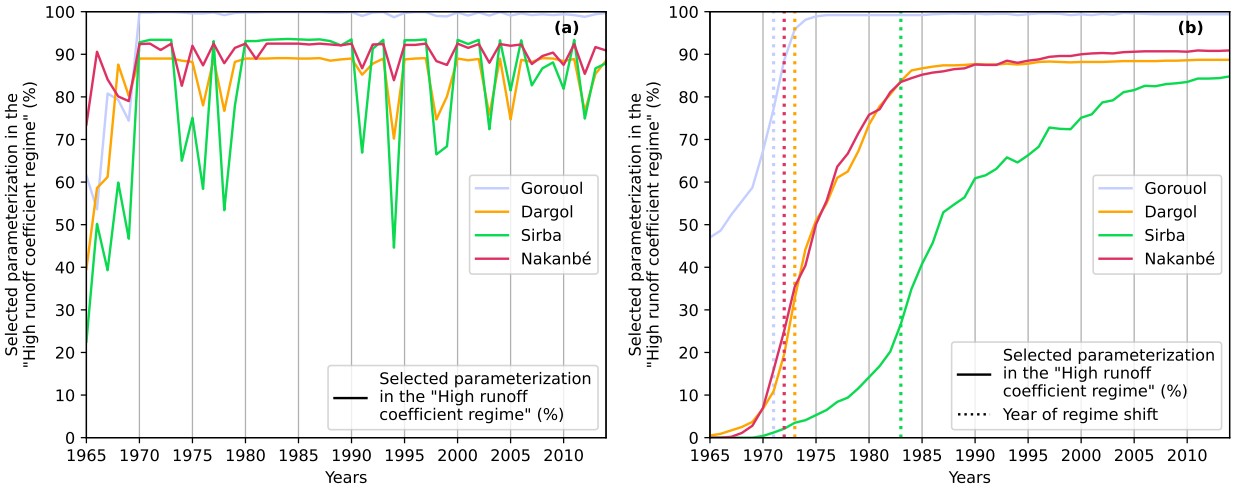

**Figure 8.** Percentage of selected parameterizations in the "High runoff coefficient regime" between 1965 and 2014. We illustrate this percentage for the four watersheds and for two definitions of regimes **(a)** regimes based on attraction basin **(b)** regimes based on a threshold.

To analyze the timing of the regime shift, we define the year of the shift as the year with the greatest number of regime shifts. Graphically, it corresponds to the year when the slope of the curve "percentage vs. time" is maximized. In practice, we compute it as the year where "% of selected parameterization in the High runoff coefficient regime(year+1)-% of selected parameterization in the High runoff coefficient regime (year-1)" is maximized. Following this definition, the regime shift occurred in 1971, 1972, 1973, 1983 for the Gorouol, Nakanbé, Dargol, and Sirba watershed, respectively (Figure 8 **b**). For the

Dargol and Nakanbé watersheds, the percentage of selected parameterizations in the "High runoff coefficient regime" increased from $0\%$ to $75\%$ in 15 years, while it took approximately 30 years for the Sirba watershed.

# 5 Discussion

The estimated timing of the regime shift depends on several choices such as i) considering the year of the shift as the year with the highest number of regime shift; ii) selecting the definition of regimes separated by a threshold; iii) deciding to calculate attractors until 4000 mm; and iv) designing the dynamical model and its external forcing. In the rest of this discussion, we will mainly discuss the design of the dynamical model.

The proposed dynamical model is a lumped model operating at the annual timescale and at the watershed scale, with precipitation as unique forcing. This deliberate choice was made for two main reasons. First, in this data-scarce region, one could hardly find the appropriate datasets back in the 1950s (i.e. before the satellite era) to realistically inform a complex, distributed model. Second, existing hydrological models poorly represent regime shifts mainly because they do not include key feedback processes (Avanzi et al., 2020; Fowler et al., 2022a). Therefore we develop a model, with limited complexity that is consistent with the available datasets and includes a representation of feedback processes, based on earlier work (Wendling et al., 2019). Although the equations of the model are not derived from physics, they describe real hydrological mechanisms, rooted in the most recent knowledge about eco-hydrology in Sub-Saharan Africa. Despite its limited complexity, the model reproduces well the increasing trend in runoff coefficient, for which it was designed. However, the model under-represents year-to-year runoff variability, and many observed values fall outside the 5%-95% quantile range of simulated runoff coefficient (Fig. 6). Compared to theoretical studies involving feedback loops e.g. (Van Nes et al., 2014), our approach goes one step further by using observations to externally force and constrain the model. The fact that our model performs better than a standard model forced by annual precipitation (App. C) suggests that this model strikes an acceptable balance between realism and predictability.

This dynamical model has a unique external forcing: annual precipitation. This strong assumption derives from Wendling et al. 2019, where the prolonged precipitation deficit (the drought) was shown sufficient to trigger the regime shift. However, runoff in the Sahel also depends on other factors. Land clearing is the main driver of anthropogenic land cover changes in a large part of the region. These changes, together with the increase of precipitation intensities, contribute to the increase runoff in the region (Séguis et al., 2004; Gal, 2016; Descroix et al., 2012, 2018; Gbohoui et al., 2021; Yonaba et al., 2021, 2023; Bennour et al., 2023). The main reason we only use precipitation in this work is to test our unusual modelling approach in the simplest configuration, i.e. with the assumption that precipitation is the main determinant of hydrological behavior at the watershed scale. Accounting for land cover changes in the model is a methodological challenge since it requires combining endogenous dynamics (driven in the model by feedback mechanisms) and external forcing, while avoiding excessive constraints on either. This issue will be addressed in subsequent model developments, where the addition of further external forcing will enhance the realism of the model and enable a more detailed assessment of the contribution of each factor to hydrological regime shifts.

To our knowledge, there is no established methodology to constrain and select an ensemble of parameterization in the context of non-autonomous dynamical models. Although intellectually more satisfying, a formal Bayesian approach to calibrate an ensemble would need a series of assumptions (on the prior, the likelihood, the dependence) whose justification is all but obvious. Therefore in this study we simply select the 1000 parameterizations with the lowest root mean square error (Sect. 3.2), and use them to analyze regime shifts.

For more than 90% of selected parameterizations, changes in the simulated runoff coefficients are induced by a regime shift for all the watersheds. However, for the Nakanbé, Dargol and Sirba watersheds, a small fraction of selected parameterizations (less than 10%) are monostable, i.e. with one unique regime and without tipping points. This small fraction is insensitive to the maximum level of precipitation (4000 mm in our methodology) used to compute attractors, as long as it stays above 1500 mm (App. B). The fact that we can obtain both monostable and bistable parameterizations suggests that the proposed model is flexible enough to simulate changes in runoff coefficients with or without tipping points.

## 6 Conclusion

In this article, we assume that the Sahelian paradox (increase in the annual runoff coefficient of Sahelian watersheds during the droughts in the '70s-'80s and after them) is a hydrological regime shift from a low runoff coefficient regime to a high regime, and ask: when did these regime shifts occur ? For the Gorouol, Nakanbé, Dargol and Sirba watershed, our results show that Sahelian watersheds shifted during the droughts of the '70s-'80s (in 1971, 1972, 1973 and 1983 respectively). These results were obtained with a parsimonious model which deliberately neglects fine-scale processes of Sahelian hydrology. It would therefore be wise to supplement this analysis with other models - with varying levels of complexity - that also allow regime shifting. These results depend on choices for our two key contributions: the dynamical model and the definition of regimes based on a threshold. Next we summarize these contributions and their potential extensions.

First, we develop a lumped dynamical model driven by precipitation that represents the runoff coefficient of a watershed. This simple model accounts for feedback analogous to the growth and death of vegetation patches. Our results show that the proposed model can reproduce the observed trend in runoff coefficient, even though the year-to-year variability is under-estimated. This model performs better than a classical hydrological model (without feedback) that fails to reproduce the observed trend. Our model only requires precipitation and runoff data. It could be used on other semi-arid regions, where runoff coefficient may have experienced a hydrological regime shift. We could also rely on this dynamical model and climate projections to identify watersheds that are likely to experience a regime shift in the future. New model developments would be needed to account for the expected intensification of climatic and anthropogenic pressures on drylands (Wang et al., 2022).

Second, we propose a novel definition of regimes for bistable parameterizations of the dynamical model. This definition makes it possible to identify regime shifts in the context of transient dynamics, where state values remain far from attractors because the model is too slow to adjust to fast changes in the external forcing. Such quantitative definition of regimes could pave the way to design attribution study for regime shifts, i.e. to assess whether past regime shifts have been made more or less likely by some specific causes, such as anthropogenic emissions.

In the future, regime shifts in hydrosystems could have direct implications for the adaptation to extreme hydrological events (flood, drought), as well as for water resources management and planning. Indeed, regime shift can lead to unexpected consequences as shown by the Sahelian paradox. The approach proposed in this study could improve the modelling and characterization of hydrological regime shifts. Better accounting for regime shifts is a major challenge for the future of hydrology.

*Author contributions.* ELR, VW, GP and CP designed the research. VW and CP designed the dynamical model. ELR performed the analysis and drafted the manuscript. CP ensured the funding acquisition. All authors discussed the results and edited the manuscript.

*Competing interests.* The authors declare that they have no conflict of interest.

*Acknowledgements.* This study was funded by the ANR (France) under contract no. ANR-20-CE01-0014-01 (TipHyc project). We kindly thank the Directorate-General for Water Resources (Ministry of Environment, Water and Sanitation) of Burkina Faso for providing the discharge dataset of the Nakanbé watershed.

## Appendix A:  Quantitative assessment of the fit for the dynamical model

In Table A1, for each watershed, we compute the root mean squared error (RMSE), bias, and Nash-Sutcliffe efficiency (NSE) of the runoff coefficient for the top 1000 parameterizations. The RMSE and bias are very low, which confirm the general good agreement between the simulations and the observations. Regarding the NSE criterion, the performances are good (e.g. Dargol) to moderate (Sirba, Gorouol and Nakanbé). Even if the model cannot simulate year-to-year runoff variability, the main objective of reproducing the trend i.e. the first-order hydrological dynamics on decadal time scale is fulfilled (Fig. 6).

|  | RMSE | Bias | NSE |
|---|---|---|---|
| Gorouol | 0.019 (0.018,0.02) | -0.0 (-0.01,0.01) | 0.46 (0.43,0.53) |
| Dargol | 0.027 (0.026,0.28) | -0.0 (-0.01,0.01) | 0.72 (0.71,0.75) |
| Sirba | 0.024 (0.022,0.025) | -0.0 (-0.01,0.01) | 0.57 (0.54,0.66) |
| Nakanbé | 0.013 (0.013,0.013) | 0.0 (-0.0,0.0) | 0.47 (0.45,0.5) |

**Table A1.** Root mean squared error (RMSE), bias, and Nash-Sutcliffe efficiency (NSE) for the four watersheds. Each score is displayed "mean (minimum, maximum)", where for instance "minimum" corresponds to the minimum score for the top 1000 selected parameterization.

## Appendix B: Sensitivity of bistable parameterization with respect to the maximum level for the external forcing

In Figure B1, we analyze the variation of the percentage of bistable parameterization (among the top 1000 selected parameterization) with respect to the maximum level of precipitation considered to compute attractors (Sect. 3.3). We observe that between 1500 mm and 4000 mm the percentage of bistable selected parameterizations is almost constant for three watersheds (Gorouol, Dargol, Nakanbé). Thus, the number of bistable selected parameterizations is insensitive to the maximum threshold,

as long as it is above 1500 mm. For the Nakanbé watershed, the percentage of bistable selected parameterizations is sensitive to the threshold (the percentage of bistability is increasing with the threshold). However, thankfully, our definition of "regime shift year" (Sect. 4.3), which does not depend on a percentage of bistable selected parameterizations that shift, implies that the sensitivity of the Nakanbé watershed will have little impact on the timing of regime shifts.

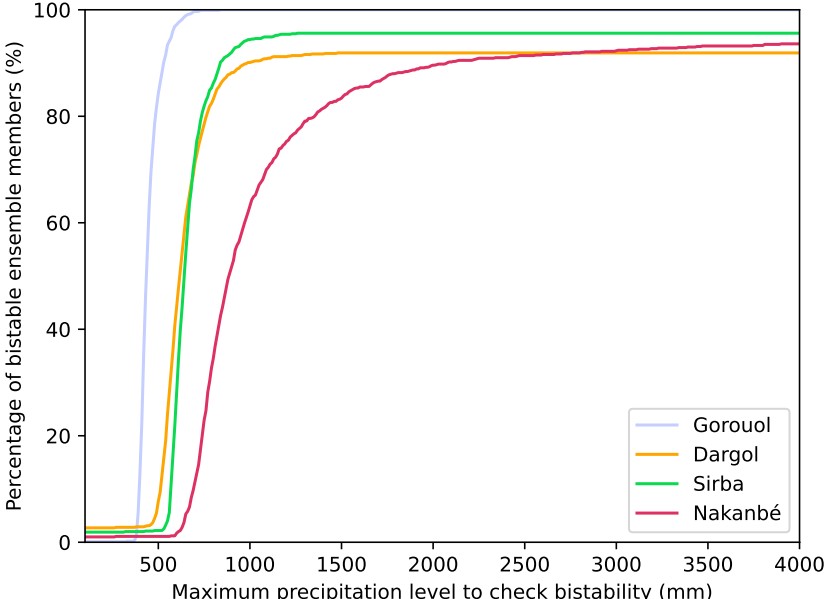

**Figure B1.** Percentage of bistable selected parameterizations with respect to the maximum precipitation level to check bistability (mm)

## Appendix C:  Benchmark against the model GR1A

335 We compare our model against the model GR1A (Mouelhi et al., 2006). This lumped bucket model with one single parameter runs at the annual time, using precipitation and potential evaporation (PET) as forcing. We use the same precipitation dataset as for our model, and annual watershed-average PET derived from ERA5. This model cannot represent feedback processes. The model was tuned with respect to the runoff coefficient for each watershed. Figure C1 shows that the simulated $K$ is over-estimated at the beginning of the period and under-estimated at the end. The GR1A model fails to capture trends in runoff

coefficients, and for the four watersheds, it performs worse than our model (Tab. C1).

|  | GR1A | our model |
|---|---|---|
| Gorouol | 0.03 | 0.02 |
| Dargol | 0.06 | 0.03 |
| Sirba | 0.04 | 0.02 |
| Nakanbé | 0.02 | 0.01 |

**Table C1.** Benchmark of our dynamical model against the GR1A model. Comparison of the root mean squared error (RMSE) for the best fit of GR1A and the mean RMSE of the ensemble simulation (our model).

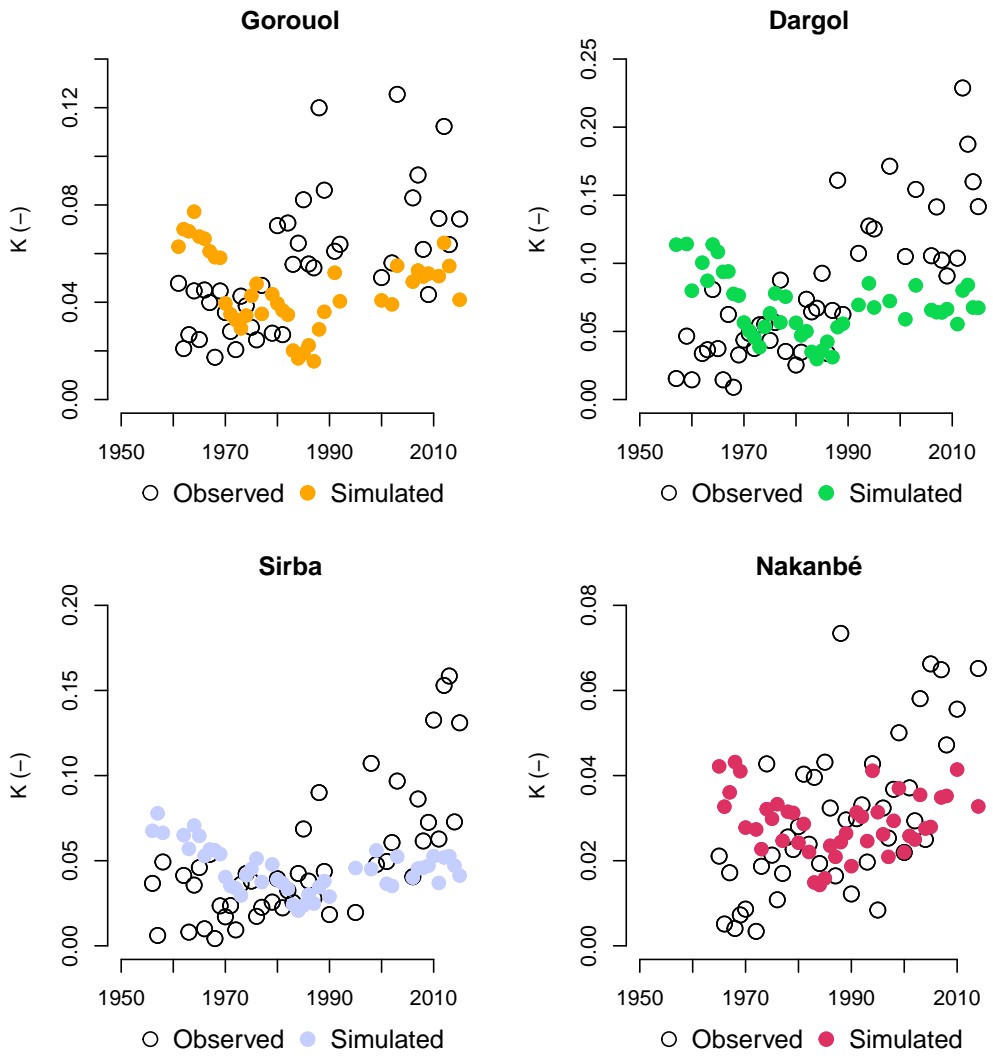

**Figure C1.** Fit of the model GR1A on the four Sahelian watersheds considered in our study. Runoff coefficient observations displayed as white dots on the Figures. Simulated runoff coefficients are displayed as colored dots.

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
