# Peer review of "Hydrological regime shifts in Sahelian watersheds: an investigation with a simple dynamical model driven by annual precipitation"

_EGUsphere, 2025_

## Author Comment (AC1)

We thank the reviewer #2 for his/her thorough comments on our manuscript. Please find below a detailed feedback to individual comments and questions

1. The introduction presents a cursory outline of existing work on hydrological regime shifts. Importantly, a shift in runoff per unit rainfall (e.g. Saft et al., 2015) is not evidence a regime shift to an alternate attractor. Doing this requires cessation of the disturbance and evidence of non-recovery, as demonstrated by Peterson et al. (2021).

Yes we completely agree that a shift in runoff per unit rainfall is not evidence of a regime shift to an alternate attractor. In the introduction of this MS, we define a regime shift as the transition from an attraction basin to an alternative attraction basin.

During the outline of existing work on hydrological regime shifts, we cite Saft et al. (2015) and Peterson et al. (2021) as hydrological studies that "rely solely on data to emphasize multiple lines of evidence that could be related to a regime shift". In the revised MS, we will remove the erroneous statement "rely solely on data". Indeed studies such as Peterson et al. (2021), do not rely on attractors and a dynamical model, but on statistical models (HMMs) and cannot be categorized as "relying solely on data".

As reminded by the reviewer, multiple lines of evidence can suggest the existence of a regime shift, i.e. the shift to an alternate attractor. These evidence include i) a shift in runoff per unit rainfall, ii) a cessation of the disturbance, and iii) an evidence of non-recovery. In the introduction of the revised MS, we will detail these multiple lines of evidence for Sahelian watershed: the cessation of the disturbance (the drought) was analyzed in Panthou et al. (2018), while the two other lines of evidence are studied in Descroix et al (2009 and 2018). These lines of evidence are analysed in a companion paper which is under review. In the revised MS, we will cite this companion paper if it gets published in the meantime. In this MS, we focus on a complementary line of evidence based on a dynamical model.

Additionally, the mechanisms for regime shifts is vague (Peterson et al., 2012), the role of forcing on regimes is not examined (Peterson et al., 2014; Peterson and Western, 2014) and informative case studies using numerical hydrological models are overlooked (Anderies, 2005; Anderies et al., 2006).

Yes, we missed these additional references from Australian hydrology. In the revised MS, we will include references to these articles in the introduction.

- (Anderies, 2005, Anderies et al., 2006) Both articles extract unstable and stable equilibrium from a minimal dynamical model applied to a catchment.

- (Peterson et al., 2012) extract attractors, repellors and attraction basin for a simple lumped groundwater model.

- (Peterson and Western, 2014, Peterson et al., 2014) This two-part article relies on an ecohydrological model to investigate the existence of multiple stable-states.

Additionally, the many efforts by others to identify the timing of shifts (Peterson et al., 2021) [...] is overlooked [...].

In the article (Peterson et al., 2021), that we read and cited in the preprint, we initially did not notice that they were also identifying the timing of shifts. But indeed, in Figure S19 and S20 in the Supplementary Material, there are two graphs where we can see the conditional state probabilities over time, and the year when the probability crosses 50%. In S19 there is a switch from normal to low state, while in S20 there is a switch from high to normal state.

In the revised MS, we will mention in the introduction that other works have previously identified the timing of shifts and cite (Peterson et al., 2021). This finding does not make our contribution outdated, since in (Peterson et al., 2021) regimes are identified statistically, while in our MS the regimes are identified with the bifurcation diagram.

Additionally, the many efforts by others to identify [...] the hydrological mechanisms is overlooked (Fowler et al., 2016, 2020; Saft et al., 2016).

Thank you for all these additional references, from Australian hydrology, that we missed.

- (Fowler et al 2016) analyzes from the literature that conceptual rainfall-runoff models are leading to poor performance when evaluated over multiyear droughts.

- (Saft et al 2016) asks whether model performance degradation is due to climate shift only or to shifts in internal catchment functioning

- (Fowler et al 2020) points out that conceptual rainfall-runoff models should be improved to account for long and slow dynamics, i.e. storage/memory effects.

In the introduction of the revised MS, we will include references to these articles.

2. The MS asks did these regime shifts occur? Given well established statistical models and code exist for this, e.g. Hidden Markov Models (Peterson et al., 2021), it is very unclear why the proposed approach is appropriate.

We agree that Hidden Markov Models (Peterson et al. 2021) could be adapted to study the timing of regime shift. However, studying regime shifts and their timing requires multiple lines of evidence. Here, we propose an original and complementary line of evidence, based on a dynamical model, which seems relevant and appropriate. Indeed, in our MS, regimes are identified from a bifurcation diagram (and therefore correspond to the classical definition of regime shift in system dynamics), while in (Peterson et al., 2021) the timing of regime shift is also investigated but regimes are identified statistically from data.

The MS may be misleading, since we do not ask whether these regime shifts occured. The occurrence of a regime shift is a starting assumption, based on early works such as Panthou et al. (2018), Gal et al. (2017) and Descroix et al. (2009, 2018). This MS rather asks: when did these regime shifts occur ? As detailed in the introduction, "for every watershed, we assume that this Sahelian hydrological paradox corresponds to a hydrological regime shift from a low to a high runoff coefficient regime, and ask: when did the shift between these two regimes occur ?".

3. The use of an ODE to identify attractors etc is interesting. The ODE developed, however, lacks a clear hydrological basis and does not draw on well established hydrological processes. Overall, it appears to be drawn from the school of ecosystem resilience that has for too long relied on toy models that are incapable of explaining observations or offering practical insights (Newton, 2016). I urge the authors to develop a model that is based on hydrological mechanisms.

We recognize that it could appear at first as developed from the school of ecosystem resilience, however as stated in the MS, the ODE is drawn from the school of ecological modeling, which can sometimes involve studies on resilience but not necessarily. This MS does not focus on resilience at all.

The co-authors of this MS have decades-long experience in tropical eco-hydrology, gained in the field and with process studies (e.g review in Lebel et al, 2009, Galle et al, 2018). Many of them are also very familiar with physically based models, as a developer or an advanced user (e.g. Casse et al, 2016, Gal et al, 2017; Getriana et al, 2017 ; Hector et al, 2018). We deliberately chose to develop a minimal model capable of reproducing the first-order processes (Scheffer and Carpenter, 2003). In this data-scarce region, the comprehensive dataset required to inform a more complex, physically-based model throughout the investigation period is not available, especially before the satellite era. Therefore our model is consistent with both the available datasets and the basic assumption of the work that a regime shift occurred and that drought was a sufficient trigger. To the best of our knowledge, no formalism capable of representing the feedbacks we introduce have been published. Following the parsimony principle of model development, we started with a simple model, aiming at improving our understanding of these overlooked interactions which drive such complex eco-hydrological systems (Sivapalan 2018).

The model has been built from the most recent state of knowledge about eco-hydrological processes in Subsaharan Africa. Although the model equations are not properly based on physics, they describe real hydrological mechanisms using response functions (e.g. Eq. 1, K vs P relationship, see also answer to Rev #1 about this equation). As described in the methodology section, most variables and parameters of the model can be assigned an hydrological meaning. Therefore, we do not agree that it is a "toy-model". Compared to theoretical studies with toy-models, where models are forced with long simulated time series of constant external forcing, and where the parameters are not really constrained, our approach goes one step beyond by confronting the model with observations. Any model with stationary internal basin properties - e.g. by neutralizing the feedback loop in our model- fail to reproduce the observations (see new data in answer to Rev #1). Our model reproduces the trends (or first-order dynamics) in the observations, based on the assumptions upon which it was built. In this sense we consider it "explains" the observations. However, like any other model, it is considered relevant until its skills degrade or one assumption is disproved. Hence we claim that our modelling framework is an adequate compromise between complexity and performance, for the emerging issue of regime shift investigations.

4. The approach for identifying the steady state regimes (i.e. attractors) and the fold points is very problematic. The MS presents an ODE but then uses time-solutions rather than bifurcation (Eq 4). Very well established analytical and numerical methods exist that estimate stable and unstable states with a forcing variable and then also the fold points, i.e. the thresholds between states. I urge the authors to look at these hydrological studies (Peterson, 2009; Peterson et al., 2012; D'Odorico and Porporato, 2004; D'Odorico et al., 2005), worked examples (Ludwig et al., 1997) and mathematical references (Kuznetsov , 2004; Dhooge et al., 2003). Using such methods, it should be possible to probabilistically quantify the regimes and thresholds with significantly more confidence.

In the MS we rely on time-solutions with two different initialisations, following the methodology used in a previous article from our group (Wendling et al. 2019). Thank you for the reference of Dhooge et al (2003), which refers to a Matlab package for numerical bifurcation analysis called MATCON. Indeed, such a package for bifurcation analysis (based on continuation method) seems more adapted to estimate stable and unstable states.

In the revised MS, we re-compute stable and unstable states using a Python package for numerical bifurcation analysis called "pycont-lint", which is also based on a continuation method. We consider this is a considerable improvement in our methodology. Such a continuation method requires starting from a known solution. In practice, we initialize it with a state value near 0 (S=0.01) for a forcing precipitation close to 0 (P=0.1). Then, one iteration later the algorithm starts following the isoline such that dS/dt = 0. We stop the continuation method when the trajectory goes beyond a precipitation level equal to 4000 mm. Finally, as a postprocessing step, we linearly interpolate the trajectory to obtain the value of stable and unstable states only for integer values of precipitation (1, …, 4000).

In the Figure below (Fig. 1), we show an updated version of the bifurcation diagram shown in the MS. We observe that the attractors are similar. Thanks to the continuation method we also have access to the unstable state, i.e. repulsor, displayed with a dotted line here.

[Figure]

Fig 1: Updated version of the bistable bifurcation diagram from the manuscript (Fig 4 a) with the novel methodology to compute attractors.

5. Section 3.2 states that the ODE is calibrated. This is not correct. The MS samples the parameter space but does not use any objective function to either reject implausible parameters (e.g. GLUE), or estimate formal likelihoods (e.g. Vrugt, 2016). Similarly, the approach cannot also be called sensitivity analysis, given the lack of quantitative evaluation against observed flow.

In the MS we do not state that "the ODE is calibrated", but that we calibrate an ensemble of parameterization. We agree with the reviewer that the word "calibration" may not be adapted to our approach. Indeed, a calibrated ensemble would ideally be selected by minimizing an objective function that evaluates the quality of an entire ensemble, so that selected parameterizations are complementary/well-combined. In our approach, this is not the case, as parameterizations are selected independently/separately.

In the revised MS, we will remove the term "calibrated ensemble", and we will rename "calibration of the model" by "selection of the best parameterizations of the model".

To the best of our knowledge, in the context of non-autonomous dynamical models, such calibration of an ensemble, e.g. based on a formal Bayesian approach, would likely be extremely difficult. Let $K = [K_{t0}, \ldots, K_t]$ be a time series of observed runoff coefficients, and $\theta$ a parameterization. To compute/estimate the posterior $p(\theta|K)$ we would need to define the prior $p(\theta)$ and the likelihood $p(K|\theta)$. Defining the joint prior distribution $p(\theta)$ would involve choosing marginal distributions for each parameter and a dependence structure between them. For most of them this dependence is unknown. Decomposing the likelihood $p(K|\theta)$ would need some assumptions. Generally, conditional independence between successive time steps is assumed, such that $p(K|\theta)$ can be decomposed as the product of $p(K_t|\theta)$. However, in our context this assumption would likely be erroneous because there is a strong temporal dependency. One solution could be to rely on a hidden Markov model, like Peterson et al. (2021), but this would likely require many additional assumptions. Although intellectually more satisfying, a formal Bayesian approach would need a series of assumptions whose justification is all but obvious. For all these reasons, we retained a simpler approach.

The claim that our approach does not use "any objective function to reject implausible parameters" suggests we have to clarify our method. In our methodology, the selection of the best parametrizations, which amounts to rejecting the implausible ones, relies on the root mean squared error between observed and simulated runoff coefficients. The mean squared error (MSE) is a standard objective function in statistics that can be interpreted as maximum likelihood estimation, when one assumes that the target distribution is Gaussian (Sect 5.5.1 of Goodfellow et al. 2015).

Additionally, the predictor (rainfall) is not independent of the predicted variable (i.e. runoff ratio) because it is used in the denominator of the runoff ratio. I urge the authors to develop a formal likelihood function for flow (not runoff ratio) and then do MCMC estimation of the parameter uncertainty.

The term "predictor" and "predicted variables" are not used in the MS, and are misleading in this context, as they refer to a statistical learning framework. Here rainfall is an external forcing for a dynamical model that simulates the runoff coefficient. For such an non-autonomous dynamical model, there is no issue that the simulated variable is dependent on the external forcing variable, with the exception that this dependence might make calibration/fitting more difficult. Here the model does not predict runoff coefficient for a given year using only rainfall of the same year. As shown in Eq. 1, the runoff coefficient for a given year depends both on rainfall P of the same year, and on the state variable S (through the variable C). The value of the state variable S does not depend only on the rainfall of the current year, but also on the previous values of S, hence on the whole rainfall history (through Eq. 3).

In many articles on hydrological regime shift, the watershed outflow is modelled because it is the key physical variable enduring a regime shift (usually from flow to low flow). However, for the considered Sahelian watersheds, we observe a hydrological paradox (assumed to be a regime shift in this MS) with respect to the runoff coefficient. Thus, this is the reason why our dynamical model focuses on modelling the dynamics of the runoff coefficient.

References:

Anderies, J. M. (2005), Minimal models and agroecological policy at the regional scale: an application to salinity problems in southeastern Australia, Regional Environmental Change, 5 (1), 1–17, doi:10.1007/s10113-004-0081-z.

Anderies, J. M., P. Ryan, and B. H. Walker (2006), Loss of resilience, crisis, and institutional change: lessons from an intensive agricultural system in southeastern Australia, Ecosystems, 9 (6), 865–878, doi:10.1007/s10021-006-0017-1.

Casse, C.,et al 2016. Model-based study of the role of rainfall and land use–land cover in the changes in the occurrence and intensity of Niger red floods in Niamey between 1953and 2012. Hydrol. Earth Syst. Sci. 20, 2841–2859. https://doi.org/10.5194/hess-20-2841-2016

Descroix, L., et al.: Spatio-temporal variability of hydrological regimes around the boundaries between Sahelian and Sudanian areas of West Africa: A synthesis, Journal of Hydrology, 375, 90–102, https://doi.org/10.1016/j.jhydrol.2008.12.012, 2009

Descroix, L., et al.: Evolution of surface hydrology in the Sahelo-Sudanian Strip: An updated review, Water (Switzerland), 10, https://doi.org/10.3390/w10060748, 2018.

Dhooge, A., W. Govaerts, and Y. A. Kuznetsov (2003), MATCONT: a MATLAB package for numerical bifurcation analysis of ODEs, ACM transactions on mathematical software, 29 (2), 141–164.

D'Odorico, P., and A. Porporato (2004), Preferential states in soil moisture and climate dynamics., Proc. Natl. Acad. Sci. U. S. A., 101 (24), 8848–8851.

D'Odorico, P., F. Laio, and L. Ridolfi (2005), Noise-induced stability in dryland plant ecosystems., Proc. Natl. Acad. Sci. U. S. A., 102 (31), 10,819–10,822, doi:10.1073/pnas.0502884102.

Fowler, K., W. Knoben, M. Peel, T. Peterson, D. Ryu, M. Saft, K.-W. Seo, and A. Western (2020), Many commonly used rainfall-runoff models lack long, slow dynamics: Implications for runoff projections, 56 (5), e2019WR025,286, doi:https://doi.org/10.1029/2019WR025286, e2019WR025286 2019WR025286.

Fowler, K. J. A., M. C. Peel, A. W. Western, L. Zhang, and T. J. Peterson (2016), Simulating runoff under changing climatic conditions: Revisiting an apparent deficiency of conceptual rainfall-runoff models, Water Resour. Res.,pp. n/a–n/a, doi:10.1002/2015WR018068.

Galle, S.,et al, 2018. AMMA-CATCH, a Critical Zone Observatory in West Africa Monitoring a Region in Transition. Vadose Zone Journal 17. https://doi.org/10.2136/vzj2018.03.0062

Gal, L., et al, 2017. The paradoxical evolution of runoff in the pastoral Sahel: analysis of the hydrological changes over the Agoufou watershed (Mali) using the KINEROS-2 model. Hydrology and Earth System Sciences 21, 4591–4613. https://doi.org/10.5194/hess-21-4591-2017

Getirana, A., et al, 2017. Streamflows over a West African Basin from the ALMIP2 model ensemble. Journal of Hydrometeorology 18, 1831–1845. https://doi.org/10.1175/jhm-d-16-0233.1

Goodfellow, I., Bengio, Y., & Courville, A. 2016. Deep learning. MIT Press, https://dl.acm.org/doi/book/10.5555/3086952

Hector, B., et al, 2018. Hydrological functioning of western African inland valleys explored with a critical zone model. Hydrology and Earth System Sciences 22, 5867–5888. https://doi.org/10.5194/hess-22-5867-2018

Kuznetsov, Y. A. (2004), Elements of applied bifurcation theory, 3rd ed., xxii, 631 pp., Springer-Verlag, New York. Ludwig, D., B. H. Walker, and C. S. Holling (1997), Sustainability, stability, and resilience, Conservation Ecology, 1 (1), http://www.consecol.org/vol1/iss1/art7/.

Lebel, T., et al, 2009. AMMA-CATCH studies in the Sahelian region of West-Africa: An overview. JOURNAL OF HYDROLOGY 375, 3–13. https://doi.org/10.1016/j.jhydrol.2009.03.020

Newton, A. C. (2016), Biodiversity risks of adopting resilience as a policy goal,Conservation Letters, 9 (5), 369–376, doi:10.1111/conl.12227. Peterson, T. J. (2009), Multiple hydrological steady states and resilience, Ph.D. thesis, Department of Civil and Environmental Engineering, The University of Melbourne, [http://repository.unimelb.edu.au/10187/8540].

Panthou, G., et al. : Rainfall intensification in tropical semi-arid regions: The Sahelian case, Environmental Research Letters, 13, https://doi.org/10.1088/1748-9326/aac334, 2018

Peterson, T. J., A. W. Western, and R. M. Argent (2012), Analytical methods for ecosystem resilience: A hydrological investigation, Water Resour. Res., 48, W10531, doi:10.1029/2012WR012150, [AGU Feature Paper].

Peterson, T. J., and A. W. Western (2014), Multiple hydrological attractors under stochastic daily forcing: 1. can multiple attractors exist?, Water Resour. Res., 50, 29933009, doi:10.1002/2012WR013003.

Peterson, T. J., A. W. Western, and R. M. Argent (2014), Multiple hydrological attractors under stochastic daily forcing: 2. can multiple attractors emerge?, Water Resour. Res., 50, 30103029, doi:10.1002/2012WR013004.

Peterson, T. J., M. Saft, M. C. Peel, and A. John (2021), Watersheds may not recover from drought, 372, 745–749, doi:10.1126/science.abd5085.

Saft, M., A. W. Western, L. Zhang, M. C. Peel, and N. J. Potter (2015), The influence of multiyear drought on the annual rainfall-runoff relation-ship: An australian perspective, Water Resour. Res., 51 (4), 2444–2463, doi: 10.1002/2014WR015348.

Saft, M., M. C. Peel, A. W. Western, J.-M. Perraud, and L. Zhang (2016), Bias in streamflow projections due to climate-induced shifts in catchment response, Geophysical Research Letters, 43 (4), 1574–1581, doi:10.1002/2015GL067326, 2015GL067326.

Scheffer, M., Carpenter, S.R., 2003. Catastrophic regime shifts in ecosystems: linking theory to observation. Trends in Ecology & Evolution 18, 648–656. https://doi.org/10.1016/j.tree.2003.09.002

Sivapalan, M., 2018. From engineering hydrology to Earth system science: milestones in the transformation of hydrologic science. Hydrology and Earth System Sciences 22, 1665–1693. https://doi.org/10.5194/hess-22-1665-2018

Vrugt, J. A. (2016), Markov chain monte carlo simulation using the {DREAM} software package: Theory, concepts, and {MATLAB} implementation, Environmental Modelling & Software, 75, 273 – 316, doi: http://doi.org/10.1016/j.envsoft.2015.08.013

Wendling, V. et al.: Drought-induced regime shift and resilience of a Sahelian ecohydrosystem, Environmental Research Letters, 14, https://doi.org/10.1088/1748-9326/ab3dde, 2019

---

## Author Comment (AC2)

Comments from Referee #1

We thank the reviewer #1 for his very thorough comments on our manuscript. Please find below a detailed feedback to individual comments and questions.

**Lines 45–50:** You introduce a model with precipitation as the *sole* external forcing. While conceptual simplicity is appreciated, the omission of known key drivers like land use changes (e.g., deforestation, cropland expansion, crust formation) and rainfall intensity is somehow concerning. Given existing knowledge on Sahelian hydrology, this simplification may lead to misleading causal inferences. The rationale for excluding these variables needs to be better justified, and at minimum, the implications of this omission must be critically discussed earlier.

The main assumption of this work is that a regime shift occurred in the Sahelian watersheds. Following Wendling et al (2019), we also assume (implicitly in our methodology) that the prolonged rainfall deficit was a sufficient driver to trigger this shift. Indeed, we introduce a model solely forced by rainfall, first to check that regime shifts can be reproduced using bifurcation diagrams, and to assess the timing of the shift.The main rationale for excluding key processes of Sahelian hydrology (land cover changes, rainfall intensity) is to test our unusual hydrological modelling approach in successive phases of increasing complexity. We start first with rainfall, which is the dominant driver of hydrological behaviour at the watershed scale, as a proof-of-concept of the approach. In the model, we note that soil crusting is implicitly accounted for with terms 1 and 2 in Eq. 3, which mimic the increase in runoff-prone areas in conditions detrimental to vegetation development, and the expansion of these areas when they are already present.

Therefore, yes, omitting land cover changes and rainfall intensity restricts the scope of our results, and any result should be understood within the framework of the underlying hypotheses and design choices. In the conclusion of the manuscript, we reiterate that "These results depend on several design choices of our two methodological contributions". In the revised manuscript, we will emphasize our main hypotheses earlier in the abstract and recall it later in the conclusion. Specifically, when we state that "the year of the regime shift was 1968, 1976, 1977, 1987 for the Gorouol, Dargol, Nakanbé and Sirba watershed, respectively." We will add afterwards that "these results should be taken with caution because they were obtained with a parsimonious model which neglect other important processes of Sahelian hydrology. It would therefore be wise to supplement this analysis with other models — with varying levels of complexity — that allow regime shifting. "

Causal inferences will be considered once our model includes all the key processes of Sahelian hydrology. Some work is underway to include more processes in the model:

1. Following an internship, there is on-going work to modify the model and to rely on daily rainfall to design annual indicators to substitute annual rainfall as forcing.

2. In early trials, we integrated vegetation as an additional constraint to calibrate an almost similar dynamical model. However, satellite observations were limited in time and did not show much temporal changes, which made calibration difficult.

3.  Including land cover changes is challenging as in this region, such changes are related both to endogeneous dynamics, as observed in the northern areas, (represented in our model with feed-back mechanisms) and to anthropogenic land clearing mostly for expanding rain-fed crop surfaces. The methodological issue is to cope with two external forcings (rainfall and land cover) simultaneously. Techniques such as nudging (to relax the model to an observed trajectory without totally constraining its dynamics) will be considered.

Overall, we are convinced that multiple lines of evidence are required to study hydrological regime shifts of Sahelian watersheds. This manuscript provides one line of evidence (with indeed some conceptual simplicity) based on a dynamical model and its bifurcation diagram. Complementary, in a companion paper under review, we also study the statistical links (without a dynamical model) between rainfall, vegetation and runoff coefficient. In the revised manuscript, we will cite this companion paper (if it is published in the meantime) to explain how the current manuscript supplements it.

Also, not all readers are familiar with the term "attraction basin", which needs explaination beforehand.

To clarify the term "attraction basin", we have rewritten the first paragraph of the introduction in the revised manuscript as follows: "Complex dynamical systems (ecosystems, climate system) can have, for certain external conditions, several attractors (stable states) towards which the system state converges depending on its initial value. The set of initial values converging to the same attractor defines an attraction basin. Classically, one regime is associated with each attraction basin (Mathias et al., 2024). A regime shift, i.e. the transition from one regime to the alternative regime, corresponds to the crossing of a tipping point: a critical value beyond which a system switches to the alternative regime, often abruptly and/or irreversibly (IPCC, Annex VII: Glossary, 2023). "

**Lines 100–110 (Eq. 1 and surrounding text):** The functional form used to relate S, P, and K is not adequately justified from a physical or empirical standpoint. For instance, the role of the parameters aa and bb in shaping the runoff response curve requires clearer explanation.

The relationship between K, S and P is a fully derivable variant of the popular SCS model (Mockus, 1972), and of the equation used by Massuel et al. (2011) to simulate runoff in the Sahel. It is also very similar to the formalism used by Anderies et al (2005, Eq. 20) to represent the dependence of runoff to rainfall and watershed-scale water retention capacity, analogous to our variable "C". This S-shape function models the main features of the rainfall-runoff relationship: no runoff is produced below a certain rainfall amount, the runoff ratio varies only little for heavy rainfall, and it increases roughly linearly for intermediate rainfall. The parameters a and b are shape factors allowing the model to adapt to a wide range of watersheds. The mathematical form of the equation is well adapted to ODE solving. Although it is not properly a physically based model, it provides a sound representation (i.e. consistent with known hydrological processes) of the dependence of the watershed runoff ratio K to the rainfall P, for a given – time-varying – infiltrability state of the watershed (C). We acknowledge that these justifications are lacking. A more detailed description of the model principles will be included in the revised manuscript.

Why not explore more physically based alternatives or benchmark this against empirical runoff-precipitation relationships?

We agree that the same kind of analyses have to be done with other types of models to support multiple lines of evidence about hydrological regime shifts. Currently available hydrological models poorly represent regime shifts (Avanzi et al, 2020, Fowler et al, 2022), mainly because they do not include the key feedback processes.

Implementing such feed-backs in existing models is a great challenge as it potentially requires revising the whole model structure, all the more so for complex physically-based models. This is why we chose a simpler, semi-empirical approach. Another important reason is that the datasets required to inform more complex, physically-based models, are not readily available and/or non-existent at the appropriate scale for the 1950s-1980s decades.

In the Appendix of the revised manuscript, we will include a benchmarking against the model GR1A Mouelhi et al. (2006), adapted to the annual time step that does not have a feed-back loop. In the Figures below, we illustrate the tuning of this model on the four watersheds. For each watershed, we tune its sole parameter X (constant over time) based on the runoff coefficient observations (Obs) displayed as white dots on the Figures. Simulated runoff coefficients are displayed as red dots. Based on these 4 graphs, we observe that this model fails to model the trend of runoff coefficients.

[Figure]

Previous modelling studies on Sahelian watersheds successfully reproduced observed runoff or runoff ratio only when land cover changes (which drive the hydrological response to rainfall) were prescribed (Gal et al, 2017, Casse et al, 2014, Yonaba et al, 2021 ). This is why we tentatively introduced a dynamical watershed state parameter with a simple model structure.

**Lines 105–110 (Eq. 2):** The formulation of the "indicator of wetness" I is intuitive, but its dependence on the parameter f introduces confusion, especially since ff also appears in Eq. 1. The choice to divide by f in this context is not well motivated—wouldn't this imply higher f leads to lower wetness, contrary to physical intuition?

We acknowledge that the current equations may be misleading. Due to their endorheic nature, most Sahelian watersheds have sub-domains that do not contribute to runoff at the outlet. In order to account for this partial contribution, we introduced $f$, defined as the fraction of the watershed which actually contributes to watershed outlet. The model only focuses on the contributing areas, which are also the areas where a regime shift can occur. In the current form of Eq 1, $K$ is the runoff ratio of the whole watershed, and the term multiplied by $f$ represents the runoff ratio of the contributing areas. Eq. 2 computes the wetness index from the runoff ratio of the contributing areas, which explains the term $K/f$.

In the revised manuscript, we will rewrite Eq. 1 as follows, where K* will be defined as the runoff ratio of the contributing areas only. Eq. 2 will be rewritten accordingly, Eq. 3 will be kept unchanged, and we will introduce Eq. 4, where $K$ is the whole watershed runoff ratio. Eq. 4 is needed to scale the runoff ratio between the contributing areas and the whole watershed, and compare with the observations. These equations are strictly equivalent to the current ones, but more easily readable.

$$K^* = k_{max} \cdot \left( \frac{P^a}{P^a + C^a} \right)^b \text{ with } C = c_{min} + S \times (c_{max} - c_{min}) \tag{1}$$

$$I = P \cdot (1 - K^*) \tag{2}$$

$$\dot{S} = \overbrace{r_g \cdot \frac{I}{I + i_g} \cdot S \cdot (1 - S)}^{(1)} - \overbrace{r_d \cdot \frac{i_d}{I + i_d} \cdot S}^{(2)} + \overbrace{\mu \cdot (1 - S)}^{(3)} \tag{3}$$

$$K = f \times K^* \tag{4}$$

**Lines 109–115 (Eq. 3):** Equation 3 includes a third term μ(1-S) that is introduced as a stabilizer. This is acceptable, but it remains ad hoc and may significantly affect long-term trajectories of the model. Please include a sensitivity analysis of this parameter or offer more detailed justification of its value and range.

The range of values of $\mu$ (2e-3 ; 5e-3, Table 1) are 2 to 3 orders of magnitude lower that the growth and decay rates $r_g$ and $r_g$, and the term $\mu(1-S)$ remains negligible as compared to the other terms of Eq. 3, except when $S$ becomes close to 0 (in this case terms 1 and 2 in Eq. 3 are also close to 0). Thus, this term cannot affect long-term trajectories, and is designed to play a role only when $S$ is close to 0 (this term avoids the trajectory to remain stuck in 0).

**Lines 115–125 (Model calibration):** While you cite equifinality to justify using an ensemble, you could improve the reproducibility of your methodology by clarifying the basis for choosing the "top 1,000" parameter sets. Why not explore a weighted ensemble or Bayesian approach to deal with parameter uncertainty more formally?

This methodology is reproduced from a previous publication (Wendling et al. 2019) where they sample 1 000 000 parametrizations and select the best 1000 (to be transparent, this publication has many co-authors in common with this article). Computational cost is the main reason for choosing the best 1000 parametrizations, as for each selected parameterization we need to compute 2000 attractors (between 1 mm and 2000 mm). However, we agree that there is no formal justification/basis for choosing the best 1000.

We discuss below the two proposed methodologies to explore uncertainty more formally.

- A weighted ensemble could be a possible extension. However, this would require an additional assumption to compute the weights for each selected parameterization. For instance, we could choose weights as being inversely proportional to the RMSE. Such an assumption would put more weights on the parameterization with small RMSE. This is not necessarily desirable. In our current approach, the choice of an ensemble without weights, i.e. the most parsimonious choice, implies the assumption that every selected parameterization is considered as equally plausible. This large variety of parameterization, which brings confidence in our results, is also enforced by the fact that the parameterizations are sampled using a Latin hypercube sampling, that ensures a relevant exploration of the parameter space.

- A formal Bayesian approach would correspond to a weighted ensemble, where the weight of each parameterization $\theta$ is the posterior $p(\theta|K)$, where $K = [K_{t0},…,K_t]$ is the time series of observed runoff coefficient. However, to the best of our knowledge, in the context of non-autonomous dynamical models, such a formal Bayesian approach would likely be extremely difficult. Indeed, to compute/estimate the posterior $p(\theta|K)$ we would need to define the priori $p(\theta)$ and the likelihood $p(K|\theta)$. Defining the joint prior distribution $p(\theta)$ would involve choosing marginal distributions for each parameter and a dependence structure between them. Decomposing the likelihood $p(K|\theta)$ would need some assumptions. Generally, conditional independence between successive time steps is assumed, such that $p(K|\theta)$ can decompose as the product of $p(K_t|\theta)$. However, in our context this assumption would likely be erroneous. Anyway, designing $p(K_t|\theta)$ and choosing hyperparameters would have been difficult. For all these reasons, we did not choose to explore a formal Bayesian approach.

**Lines 125–135:** You define bistability based on attractor separation, but the use of arbitrary thresholds like 2000 mm precipitation and 10,000-year simulations raises concerns. These choices might significantly affect the classification of parameter sets. You should evaluate how sensitive the bistability classification is to these design choices.

Following a suggestion from the reviewer #2, in the revised manuscript we compute attractors with a more advanced methodology, a method based on numerical continuation, which removes the arbitrary choice of 10,000-year simulations. Specifically, we rely on a Python package for numerical bifurcation analysis called "pycont-lint" (a sub-package of PyDSTool). This method requires starting from a known solution. In practice, we initialize it with a state value near 0 (S=0.01) for a forcing precipitation close to 0 (P=0.1). Then, one iteration later the algorithm starts following the isoline such that dS/dt = 0. Note that we have tested the sensitivity of the algorithm to the choice of initialization. It appears that the initial value of S has no effect on the first iteration as long as S remains below 0.1.. The iterations of the continuation method are stopped when the trajectory goes beyond a precipitation level equal to 4000 mm. Finally, as a postprocessing step, we linearly interpolate the trajectory to obtain the stable and unstable states for integer values of external forcings, i.e. precipitation equal to 1 mm, …, 4000 mm.

In the Figure below (Fig. 1), we show an updated version of the bistable bifurcation diagram from the manuscript. We observe that attractors are similar. However, thanks to the continuation method we also have access to the unstable states (repulsors) displayed with a dotted line. In the revised manuscript, for clarity, we will only show the bistable bifurcation diagram between 1 mm and 2000 mm (even though the continuation is run until 4000 mm).

[Figure]

Fig 1: Updated version of the bistable bifurcation diagram from the manuscript (Fig 4 a) with the novel methodology to compute attractors.

Next, we assess the sensitivity of bistability to the maximum threshold to compute attractors (2000 mm in the submitted manuscript, 4000 mm in the revised manuscript). Specifically, we check the percentage of bistability for other thresholds taken every 10 mm.

In the revised manuscript, we will define the attractors with the most advanced methodology (as explained above). In this case, in the Figure below (Fig. 2), we observe that between 1500 mm and 4000 mm the percentage of bistable ensemble members is almost constant for three watersheds (Gorouol, Dargol, Nakanbé). Thus, the number of bistable ensemble members is insensitive to the maximum threshold, as long as it is above 1500 mm. For the Nakanbé watershed, the percentage of bistable ensemble members is sensitive to the threshold (the percentage of bistability is increasing with the threshold). However, thankfully, our new definition of "regime shift year" (see below our answer to **Lines 180–190 and Figure 7**), which does not depend anymore on a percentage of bistable ensemble members that shift, implies that the sensitivity of the Nakanbé watershed will have little impact impact of the main conclusion of this work (dates of regime shifts).

[Figure]

Fig 2: Updated version of Figure 7 from the manuscript with the novel methodology to compute attractors.

**Lines 145–155 (Eq. 4 and definition of regime):** The operationalization of regime shifts via S=(↑S+↓S)/2 is a critical assumption. This midpoint criterion might not capture the actual dynamics of transition in transient regimes. Alternative definitions (e.g. basins of attraction) or at least a justification for this heuristic are needed.

Yes, alternative definitions of regimes can be imagined. Following the vocabulary from Mathias et al. (2024), our definition based on a threshold corresponds to the "definition of regime based on a norm", where regimes are separated by an horizontal threshold. Another possibility is to rely on a "definition of regimes based on attraction basin". The reason why we choose the first definition is that it is more intuitive: any state value S can be sorted directly in the low or high regime by comparing it with some threshold. On the other hand, with a definition of regimes based on attraction basins, a state S cannot be directly sorted in one regime, as the definition of regimes depends also on the precipitation P. Thus, the same state S can switch regimes just because the external forcing precipitation has changed. Therefore, with the definition based on attraction basins, we would expect that the trajectory switches more between low and high regimes.

Thanks to the more advanced methodology to compute attractors, we have access to repulsors (see previous answer) and therefore we can define regimes based on basins of attraction. In the Figure below (Fig. 3), we show the evolution of the percentage of ensemble members in the high regime. In this case, we find that the year of the regime shift (first year where more than 50% of ensemble members are in the high regimes) would be before the drought (and often before 1965). This is due to the fact that for every watershed many ensemble members are in the high regime in 1965.

[Figure]

Fig 3: Updated version of Figure 7 from the manuscript with the novel methodology to compute attractors and using a definition of regimes based on attraction basin.

In the revised manuscript, we will keep the initial definition of regimes for two reasons: it leads to less "switching between low and high regimes" (which seems more plausible) and it is more interpretable as any state value has a single regime (whereas, with regimes defined by attraction basin, some state value can sometimes have two possible regimes depending on the associated value of precipitation P). We will add an Appendix showing the results for the definition of regimes based on attraction basins.

**Figure 5 / Lines 155–170:** The mismatch between observed runoff coefficients and the simulated ensemble spread—especially for the Sirba and Nakanbé basins—is troubling. It casts doubt on whether the model can adequately capture year-to-year variability or nonlinear transitions. Please provide a quantitative assessment of performance beyond RMSE (e.g., Nash-Sutcliffe efficiency, bias).

We acknowledge that the model cannot simulate year-to-year variability. It has not been designed for that purpose, but rather to restitute decadal dynamics as observations show that the non-linear transitions responsible from the regime shifts operate at such time scales.

In the revised manuscript, we will include a Table in the Supplementary with the quantitative assessment as advised by the reviewer. For each watershed, we computed the RMSE, bias, and Nash-Sutcliffe efficiency (NSE) for the top 1000 parameterizations. The fit is deemed as "good" for Dargol Kakassi (all NSE are larger than 0.7), "satisfactory" for Sirba (all NSE larger than 0.54), and "unsatisfactory" for Gorouol (NSE are between 0.43 and 0.53) and Nakanbe (NSE are between 0.45 and 0.5). For these two latter cases, the absolute bias remains below 0.01. These measures are consistent since the variance of the simulated values is lower than that of the observations, hence leading in some cases to poor model performance regarding NSE, while the simulated trajectory does minimise the bias. In the manuscript, we acknowledge that the model cannot simulate year-to-year variability, mainly because forcing the model with annual rainfall does not account for the non linear dependency of runoff to rainfall at shorter time scales, and because other factors such as anthropogenic land cover changes have been omitted. This was well known from the beginning, but it did not affect the main objective of reproducing the first-order (i.e. on decadal time scales) hydrological dynamics..

**Lines 180–190 and Figure 7:** The definition of the "regime shift year" as the first time when more than 50% of ensemble members enter the high regime seems arbitrary. Why not use a probabilistic or statistical breakpoint analysis? The current criterion could lead to inconsistencies in estimating regime shift timing, as seen in the Gorouol case.

Thanks to the reviewer #2, we realized that our definition of "regime shift year" has already been proposed in the literature, see. Figure S19 and S20 in the Supplementary Material of Peterson et al. 2021. However, yes in the Gorouol case the current criterion/definition of the "regime shift year" can lead to inconsistencies.

Probabilistic or statistical breakpoint analysis does not seem to be adapted to our context. To apply such analysis, we could replace every ensemble member by 0 if the shift did not occur and by 1 if the shift occurred, and then apply some breakpoint analysis test. However, this definition and the results would not be really interpretable.

Following the advice of the reviewer (to avoid inconsistencies as seen in the Gorouol case) in the revised manuscript we propose a simpler definition of the "regime shift year": the year when the number of regime shifts is maximized. Visually, it corresponds to the year where the slope of the percentage of ensemble members in the "High regime" is maximized. In practice, we find the year where "percentage(year+1)-percentage(year-1)" is maximized.

In the Figures below (Fig. 4, Fig. 5), we compare the old definition of "regime shift year" with the new definition in two cases: with the attractors of the submitted manuscript, with the attractors of the revised manuscript.

- Attractors of the submitted manuscript. With the new definition of "regime shift year", we find that the shift occurred in 1972, 1973, 1984 for Nakanbe, Dargol, Sirba, respectively. For Gorouol the shift was found in 1971

[Figure]

Fig 4: Updated version of Figure 7 from the manuscript, with the attractors from the submitted manuscript, to compare the old definition of "regime shift year" (vertical dashed lines) with the new definition (vertical dotted lines)

- Attractors of the revised manuscript. With the new definition of "regime shift year", we find that the shift occurred in 1972, 1973, 1983 for Nakanbe, Dargol, Sirba, respectively. For Gorouol the shift was found in 1971.

[Figure]

Fig 5: Updated version of Figure 7 from the manuscript, with the attractors of the revised manuscript, to compare the old definition of "regime shift year" (vertical dashed lines) with the new definition (vertical dotted lines)

**Lines 200–220 (Discussion):** You rightly acknowledge that the model underrepresents interannual variability and that precipitation alone is insufficient. However, this admission seems to undercut the core claim that the model can meaningfully identify regime shifts. This contradiction should be addressed more transparently. Can regime shifts truly be inferred from such a limited model?

We agree that the phrase may be misleading. We meant that the model under-represents year-to-year runoff variability, yet it well captures the decadal – low-frequency – variability, which is the time scale of both the drought signal and of the emergence of the regime shift. We will clarify this point in the revised manuscript.

Our starting assumption is that the Sahelian paradox corresponds to a regime shift. Here, we present a model that can reproduce a regime shift: we calibrate the model using observations and find that most parametrizations show a trajectory of the state variable that corresponds to a regime shift. It is likely that many other models (including much simpler models) could reproduce this regime shift. However, we think that the proposed model strikes a good balance between having some complexity and reproducing the regime shift.

**Lines 205–210 (Gorouol case):** The early regime shift in the Gorouol basin (before observed droughts) is indeed counterintuitive. It may reflect model artefacts from initialization, especially since 40% of ensemble members already start in the "high" regime. This undermines the claim of detecting shifts dynamically. Please explore whether this result is robust or an artefact of initial conditions.

Yes, in the manuscript, the early regime shift likely reflects either i) initialization issues and/or ii) unadapted definition of "regime shift year". Exploring other initial conditions for Gorouol seem out of reach in terms of computation, as for a given initialization the whole workflow (solving the trajectory for 1000000 sampled parameterization, selection of the ensemble with 1000 parameterizations, estimating the attractors for the whole ensemble) takes about a week on a parallelized cluster. This is why methodologically we did the most parsimonious/simple choice of initialization: the initial state value S is computed with the first observed runoff coefficient and Eq. 1. In the revised manuscript, to fix this Gorouol case, we will follow your advice from an earlier comment and adapt the definition of "regime shift year" so that it does not account for ensemble members that already start in the "high" regime.

In practice, the early shift in the Gorouol watershed is consistent with field observations, which suggest that the most sensitive areas (shallow sandy soil over rocky/clayey substratum) have been eroded very soon after the beginning of the drought, in the early 70s, whereas it occurred later on the other watersheds, see Nguyen (2015). These details will be added to the revised manuscript.

**Lines 215–220:** The interpretation of monostable vs. bistable ensemble members is important, but underdeveloped. If 10% of simulations do not undergo regime shifts, does this reflect real watershed variability or model limitations? Some exploration of this heterogeneity would enrich the discussion.

In the submitted manuscript Gorouol, Dargol and Sirba had only bistable ensemble members, while Nakanbé had 10% of monostable ensemble members. With the novel definition of attractors, this ratio is 6.4 %, and monostable members (4.4% and 8.1%) are also detected for Sirba and Dargol, respectively. For Gorouol, this percentage equals 0%.

We hypothesize that this difference is due to the fact that our old definition of attractors may have been wrongly detecting some bistability, e.g. upper and lower attractors that were estimated as different even though in practice they corresponded to the same attractor.

A monostable ensemble member implies that the watershed did not undergo a regime shift. As stated in the manuscript, this result first shows that the proposed model can simulate changes in runoff coefficients with or without regime shift. However, monostable members represent less than 10 % of the ensemble . This result could be related to real watershed variability, but is indeed more likely related to the main model limitations (Land cover changes and daily rainfall intensity not accounted for). .

**A minor and general comment: t**he writing is generally clear, but at times overly dense with jargon. Consider simplifying key explanations, especially around dynamical systems concepts, to enhance accessibility for a broader hydrological audience. Also, Figures (in general) are informative, though Figures 4 and 6 could benefit from clearer legends and a brief description of axis choices (e.g. why is S bounded between 0–0.7?).

In the revised manuscript, following an early comment, we simplify the first paragraph of the introduction and explain the term "attraction basin". Before this preprint, this manuscript already underwent several rounds of revisions to ensure that it is accessible for a hydrological audience: many co-authors, with hydrological backgrounds, provided feedback in order to help the readability around dynamical systems concepts. Please do not hesitate to pinpoint us sentences where concepts remain unclear, we will try to fix that. Finally, in the revised manuscript, we will add such a brief description of axis choices for Figures 4 and 6.

References:

Anderies, J. M. (2005), Minimal models and agroecological policy at the regional scale: an application to salinity problems in southeastern Australia, Regional Environmental Change, 5 (1), 1–17, doi:10.1007/s10113-004-0081-z.

Avanzi, F., et al.: Climate elasticity of evapotranspiration shifts the water balance of Mediterranean climates during multi-year droughts, Hydrology and Earth System Sciences, 24, 4317–4337, https://doi.org/10.5194/hess-24-4317-2020, 2020

Cam Chi Nguyen, 2015, Dynamique, structure et production de la végétation du Gourma (Sahel, Mali) en relation avec les sols, l'occupation des sols et les systèmes hydriques : étude de télédétection à haute et moyenne résolution. PhD thesis, University of Toulouse (France). https://theses.fr/2015TOU30143

Casse, C.,et al 2016. Model-based study of the role of rainfall and land use–land cover in the changes in the occurrence and intensity of Niger red floods in Niamey between 1953and 2012. Hydrol. Earth Syst. Sci. 20, 2841–2859. https://doi.org/10.5194/hess-20-2841-2016

Fowler, K., et al (2022).: Hydrological Shifts Threaten Water Resources, https://doi.org/10.1029/2021WR031210, 2022a

Gal, L., et al, 2017. The paradoxical evolution of runoff in the pastoral Sahel: analysis of the hydrological changes over the Agoufou watershed (Mali) using the KINEROS-2 model. Hydrology and Earth System Sciences 21, 4591–4613. https://doi.org/10.5194/hess-21-4591-2017

IPCC: Annex VII: Glossary, in: Climate Change 2021 – The Physical Science Basis, pp. 2215–2256, Cambridge University Press, https://doi.org/10.1017/9781009157896.022, 2023.

Massuel, S., et al.: Modélisation intégrée surface-souterrain dans un contexte de hausse des réserves d'un aquifère régional Sahélien, Hydrological Sciences Journal, 56, 1242–1264, https://doi.org/10.1080/02626667.2011.609171, 2011

Mathias, J. D., et al..: From tipping point to tipping set: Extending the concept of regime shift to uncertain dynamics for real-world applications, Ecological Modelling, 496, https://doi.org/10.1016/j.ecolmodel.2024.110801, 2024

Mockus, V.: Estimation of Direct Runoff from Storm Rainfall., Tech. Rep. 210-VI-NEH, US Department of Agriculture, 1972

Mouelhi, S., et al., 2006. Linking stream flow to rainfall at the annual time step: The Manabe bucket model revisited. Journal of Hydrology, 328 (1), 283-296. https://www.sciencedirect.com/science/article/pii/S0022169406000023

Wendling, V. et al.: Drought-induced regime shift and resilience of a Sahelian ecohydrosystem, Environmental Research Letters, 14, https://doi.org/10.1088/1748-9326/ab3dde, 2019

Yonaba, R., et al. (2021): A dynamic land use/land cover input helps in picturing the Sahelian paradox: Assessing variability and attribution of changes in surface runoff in a Sahelian watershed, Science of the Total Environment, 757, 143 792, https://doi.org/10.1016/j.scitotenv.2020.143792, 2021